# Deciphering deep-sea chemosynthetic symbiosis by single-nucleus RNA-sequencing

Hao Wang[1,2,3,4]*†, Kai He[5]†, Huan Zhang[1]†, Quanyong Zhang[6]†, Lei Cao[1], Jing Li[7], Zhaoshan Zhong[1], Hao Chen[1], Li Zhou[1], Chao Lian[1], Minxiao Wang[1], Kai Chen[6], Pei-Yuan Qian[3,4]*, Chaolun Li[1,7,8]*

[1]Center of Deep-Sea Research, Institute of Oceanology, Chinese Academy of Sciences, Qingdao, China; [2]Laboratory for Marine Biology and Biotechnology, Qingdao Marine Science and Technology Center, Laoshan Laboratory, Qingdao, China; [3]Southern Marine Science and Engineering Guangdong Laboratory (Guangzhou), Guangzhou, China; [4]Department of Ocean Science, Hong Kong University of Science and Technology, Hong Kong, China; [5]Key Laboratory of Conservation and Application in Biodiversity of South China, School of Life Sciences, Guangzhou University, Guangzhou, China; [6]State Key Laboratory of Primate Biomedical Research, Institute of Primate Translational Medicine, Kunming University of Science and Technology, Kunming, China; [7]South China Sea Institute of Oceanology, Chinese Academy of Sciences, Guangzhou, China; [8]University of Chinese Academy of Sciences, Beijing, China

*For correspondence:
haowang@qdio.ac.cn (HW);
boqianpy@ust.hk (P-YQ);
lcl@qdio.ac.cn (CL)

†These authors contributed equally to this work

Competing interest: The authors declare that no competing interests exist.

## eLife assessment

This study provides an **important** cell-type atlas of the gill of the mussel *Gigantidas platifrons* using a single-nucleus RNA-seq dataset, a resource for the community of scientists studying deep-sea physiology and metabolism and intracellular host–symbiont relationships. The evidence supporting the conclusions is **convincing** with high-quality single-nucleus RNA sequencing and transplant experiments. This work will be of broad relevance for scientists interested in host–symbiont relationships across ecosystems.

**Abstract** Bathymodioline mussels dominate deep-sea methane seep and hydrothermal vent habitats and obtain nutrients and energy primarily through chemosynthetic endosymbiotic bacteria in the bacteriocytes of their gill. However, the molecular mechanisms that orchestrate mussel host–symbiont interactions remain unclear. Here, we constructed a comprehensive cell atlas of the gill in the mussel *Gigantidas platifrons* from the South China Sea methane seeps (1100 m depth) using single-nucleus RNA-sequencing (snRNA-seq) and whole-mount in situ hybridisation. We identified 13 types of cells, including three previously unknown ones, and uncovered unknown tissue heterogeneity. Every cell type has a designated function in supporting the gill's structure and function, creating an optimal environment for chemosynthesis, and effectively acquiring nutrients from the endosymbiotic bacteria. Analysis of snRNA-seq of in situ transplanted mussels clearly showed the shifts in cell state in response to environmental oscillations. Our findings provide insight into the principles of host–symbiont interaction and the bivalves' environmental adaption mechanisms.

## Introduction

Mutualistic interactions between multicellular animals and their microbiota play a fundamental role in animals' adaptation, ecology, and evolution (*Kremer et al., 2013*; *Bang et al., 2018*). By associating with symbionts, host animals benefit from the metabolic capabilities of their symbionts and gain fitness advantages that allow them to thrive in habitats they could not live in on their own (*Kremer et al., 2013*; *Franke et al., 2021*). Prime examples of such symbioses are bathymodioline mussels and gammaproteobacteria endosymbionts (*Dubilier et al., 2008*). The bathymodioline mussels occur worldwide at deep-sea chemosynthetic ecosystems, such as cold seeps, hydrothermal vents, and whale falls (*Dubilier et al., 2008*). The host mussel acquired sulphur-oxidising (SOX) and/or methane-oxidising (MOX) symbiont through horizontal transmission at their early life stage (*Franke et al., 2021*). Their symbionts, which are hosted in a specialised gill epithelial cell, namely, the bacteriocytes (*Sogin et al., 2021*; *Xu et al., 2019*), utilise the chemical energy from the reduced chemical compounds such as $CH_4$ and $H_2S$ released from cold seeps or hydrothermal vents to fix carbon and turn into carbon source for the host mussel (*DeChaine and Cavanaugh, 2006*; *Fujiwara et al., 2000*). The ecological success of the bathymodioline symbioses is apparent: the bathymodiolin mussels are often among the most dominant species in the deep-sea chemosynthetic ecosystems (*Kiel, 2010*; *Vrijenhoek, 2010*). Thus, it is critical to know how the bathymodioline mussels interact with the symbionts and maintain the stability and efficiency of the symbiosis.

The gill structure of bathymodiolin mussels has undergone remarkable adaptations at the molecular, cellular, and tissue levels to support their deep-sea chemosynthetic lifestyle (*Halary et al., 2008*; *Zheng et al., 2017*). Compared to shallow-water mussels (*Fiala-Médioni et al., 1986*), the gill filaments of bathymodiolin mussels have enlarged surfaces, allowing them to hold more symbionts per filament (*Figure 1—figure supplement 1*). This adaption requires not only novel cellular and molecular mechanisms to maintain and support the enlarged gill filament structure, but also a strong ciliary ventilation system to pump vent/seep fluid to fuel the symbiotic bacteria. Previous studies based on whole-genome sequencing and bulk RNA-seq projects have shown that genes in the categories of nutrient transporters, lysosomal proteins, and immune receptors are either expanded in the host mussel's genome or upregulated in the gill (*Zheng et al., 2017*; *Wong et al., 2015*; *Sun et al., 2017*; *Bettencourt et al., 2017*; *Barros et al., 2015*), suggesting that these genes are involved in the host–symbiont interaction. Though providing deep molecular insights, these studies mainly used homogenised tissues that average genes' expression levels amongst different cell types and eliminate cell and gene spatial distribution information (*Wang et al., 2021*; *Chen et al., 2015a*; *Hwang et al., 2018*; *Saliba et al., 2014*). In addition, the broad expression and function of potentially 'symbiosis-related' proteins also greatly limited data interpretation. Therefore, a systemic atlas of gill cell types and the descriptions of cell-type-specific gene expressional profiles are warranted to a better understanding of the host–symbiont interaction and environmental adaptation mechanisms of the bathymodioline symbiosis.

In recent years, single-cell/nucleus RNA-sequencing (sc/sn RNA-seq) technologies have become one of the preferred methods for investigating the composition of complex tissue at the transcriptional level (*Chen et al., 2018*). snRNA-seq has several advantages, such as compatibility with frozen samples, elimination of dissociation-induced transcriptional stress responses, and reduced dissociation bias (*Wu et al., 2019*). These advantages are significant for deep-sea and other ecological studies because cell and molecular biology facilities are not available in the field. To examine the cellular and molecular mechanisms of the environmental adaptations and host–symbiont interactions in bathymodioline mussels, we conducted an snRNA-seq-based transcriptomic study. We analysed the gill symbiosis in the dominate deep-sea mussels inhabiting the F-site cold seep *Gigantidas platifrons* (*Feng et al., 2015*), which hosts a single MOX endosymbiont population that consists of only one 16S rRNA phylotype. Our work provides a proof-of-principle for environmental adaptation mechanisms study in non-traditional but ecologically important organisms with snRNA-seq technologies.

## Results and discussion

### *G. platifrons* deep-sea in situ transplant experiment and single-cell transcriptomic sequencing

We conducted a *G. platifrons* deep-sea in situ transplant experiment at the 'F-site' cold seep (~1117 m depth) and retrieved three groups of samples (*Figure 1A*), as follows: the 'Fanmao' (meaning prosperous) group, which comprised the mussels collected from methane-rich (~40 μM) site of the cold seep, where the animals thrived; the 'starvation' group, which comprised the mussels collected from the methane-rich site and then moved to a methane-limited (~0.054 μM) starvation site for 14 days before retrieval; and the 'reconstitution' group, which comprised the methane-rich site mussels moved to the 'starvation' site for 11 days and then moved back to the methane-rich site for another 3 days.

We next performed snRNA-seq of the gill posterior tip of the three groups of samples using the microwell-based BD Rhapsody platform Single-Cell Analysis System (*Figure 1A*). After quality control, we obtained 9717 (Fanmao), 21,614 (starvation), and 28,928 (reconstitution) high-quality single-nuclei transcriptomic data.

### Cell atlas of *G. platifrons* gill

To unravel the intricate cellular composition, we utilised a reciprocal PCA (RPCA) strategy to project cells in the three samples onto the same space based on conserved expressed genes among them (*Kharchenko et al., 2014*). This strategy maximises the number of cells per cluster regardless of the organism's state and therefore maximises genes per cluster (*Elyanow et al., 2020*).

Through the implementation of Seurat, we revealed 14 cell clusters, each was associated with a set of marker genes (*Figure 1B–D*; *Hao et al., 2021*; *Stuart et al., 2019*). Given the limited availability of canonical marker genes for *G. platifrons* and molluscs in general, we undertook a meticulous approach to characterise each cell cluster. This involved (1) examining the cluster marker genes' functions, (2) identifying the expression pattern of cluster marker gene using whole-mount in situ hybridisation (WISH) and/or double fluorescent in situ hybridisation (FISH) analyses, and (3) conducting scanning electron microscopy (SEM) and transmission electron microscopy (TEM) analyses. We successfully identified and characterised 13 of the 14 cell clusters, which could be categorised into four major groups, including (1) the supportive cells, (2) the ciliary cells, (3) the proliferation cells, and (4) the bacteriocytes. One cell cluster remained ambiguous. To assess the robustness of each cell cluster, we employed a bootstrap sampling and clustering algorithm examining the similarity among clusters and obtained strong support for all clusters through a combined analysis of the three samples (*Figure 1—figure supplement 2A*). Furthermore, when examining each sample individually, we found that the majority of clusters demonstrated robust support, with the exception of the three ciliary cell clusters, which showed overlaps of assignment probabilities among them (*Figure 1—figure supplement 2B–D*). These three cell types are derived from a same precursor and exhibited relatively lower numbers of nuclei, resulting in a reduced availability of genes per cell type, which is particularly true for the food grove ciliary cell (*Supplementary file 1a*). By integrating the spatial and functional annotations of these cell clusters, we gained insight into their collective efforts in supporting the symbiotic relationship and maintaining the high-efficiency chemosynthetic system within the gill tissue.

#### Supportive cells

We have identified four supportive cell types, namely, inter lamina cells basal membrane cells (BMC)1, BMC2, and mucus cells (*Figure 2A–C*).

The inter lamina cells, marked by high expression of fibrillar-forming collagens, were the cells located between the two layers of basal membranes (*Figure 2D and E*). These cells were previously identified as amoeboid cells (*Fiala-Médioni et al., 1986*), and their function has not been explored. The inter lamina cells were densely distributed around the food groove and at the rim of the gill filament. While on the middle part of the gill filament, the inter lamina cells distributed in parallel rows. It might help connect the two sheets of basal membranes and maintain the spatial integrity of the gill filament. In addition, we observed the enrichment of ribosomal proteins involved in protein synthesis (*Supplementary file 1b*), indicating a high metabolic rate (*Petibon et al., 2021*) of inter lamina cells. This implies that the inter lamina cells may process the nutrients acquired from the symbiont.

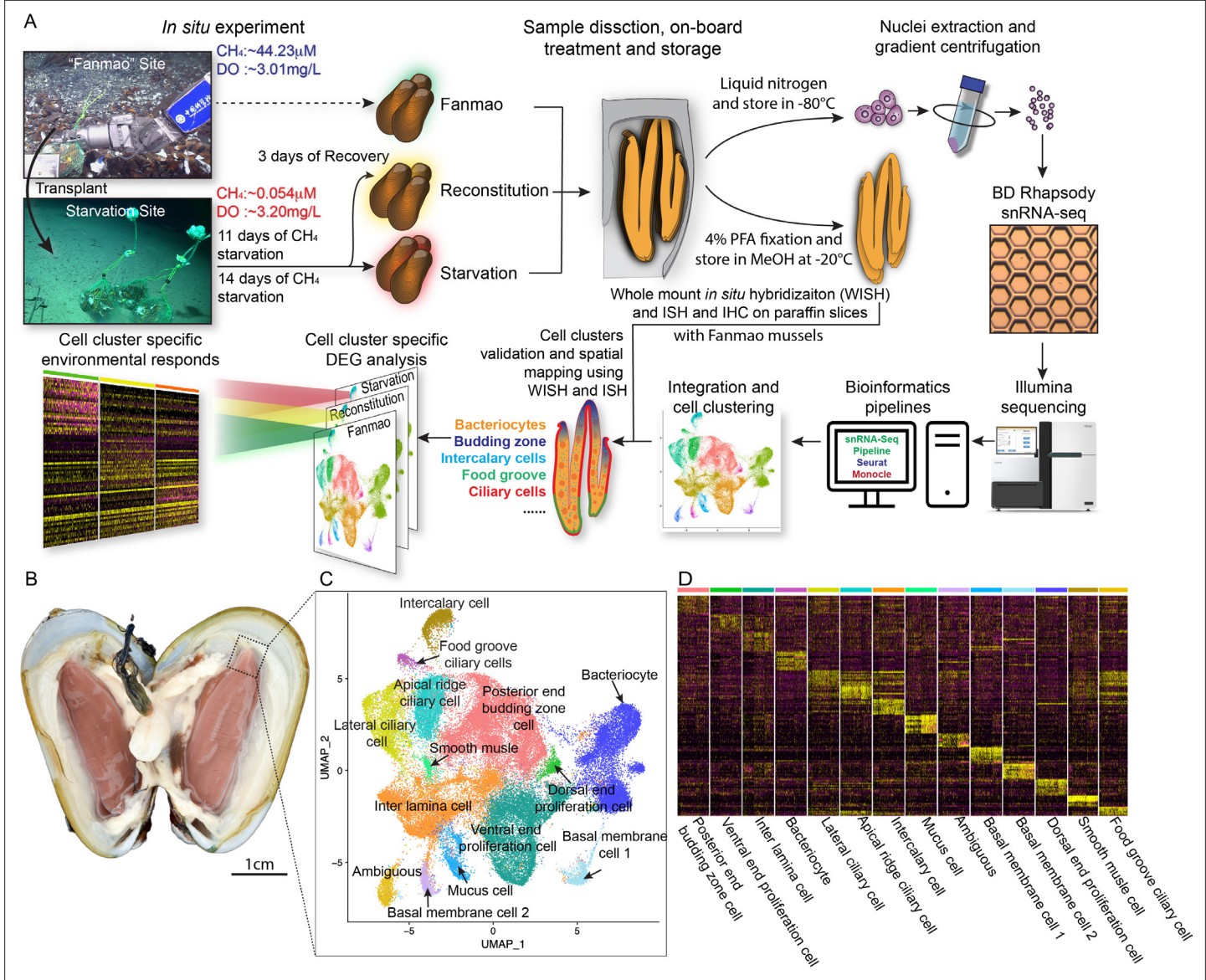

**Figure 1.** Identification of 14 cell types in the gill of deep-sea symbiotic mussel *Gigantidas platifrons*. (**A**) Overall experimental scheme describing the deep-sea in situ transplant experiment, the sample preparation procedures, and the single-cell analysis and validation pipeline. Three *G. platifrons* samples were included in the present study: 'Fanmao,' starvation, and reconstitution. The cell nucleus was extracted from each sample, which included a pool of gill posterior tip of three mussels. The snRNA-seq libraries were constructed according to the BD Rhapsody single-nuclei 3′ protocol. Cell population-specific markers were validated by whole-mount in situ hybridisation (WISH) and in situ hybridisation (ISH). (**B**) The image shows the posterior end of the gill of *G. platifrons*. (**C**) Uniform Manifold Approximation and Projection (UMAP) representation of *G. platifrons* gill single cells. Cell clusters are coloured and distinctively labelled. (**D**) Heat map profile of markers in each cluster. The colour gradient represents the expression level of every single cell.

The online version of this article includes the following figure supplement(s) for figure 1:

**Figure supplement 1.** Image showing the gill tissues of the coastal shallow-water mussel *Modiolus philippinarum* and deep-sea chemosynthetic mussel *Gigantidas platifrons*.

**Figure supplement 2.** Heat maps of co-assignment probabilities of bootstrap sampling estimated for recognised clusters using cells in all three samples (**A**) and using cells in individual sample of Fanmao (**B**), starvation, (**C**) and reconstitution (**D**).

**Figure supplement 3.** Uniform Manifold Approximation and Projection (UMAP) plots showing distribution patterns of 14 clusters in Fanmao, starvation, and reconstitution states.

**Figure supplement 4.** Stacked bar plots showing the percentages of cells per cluster in each sample (Fanmao, starvation and reconstitution).

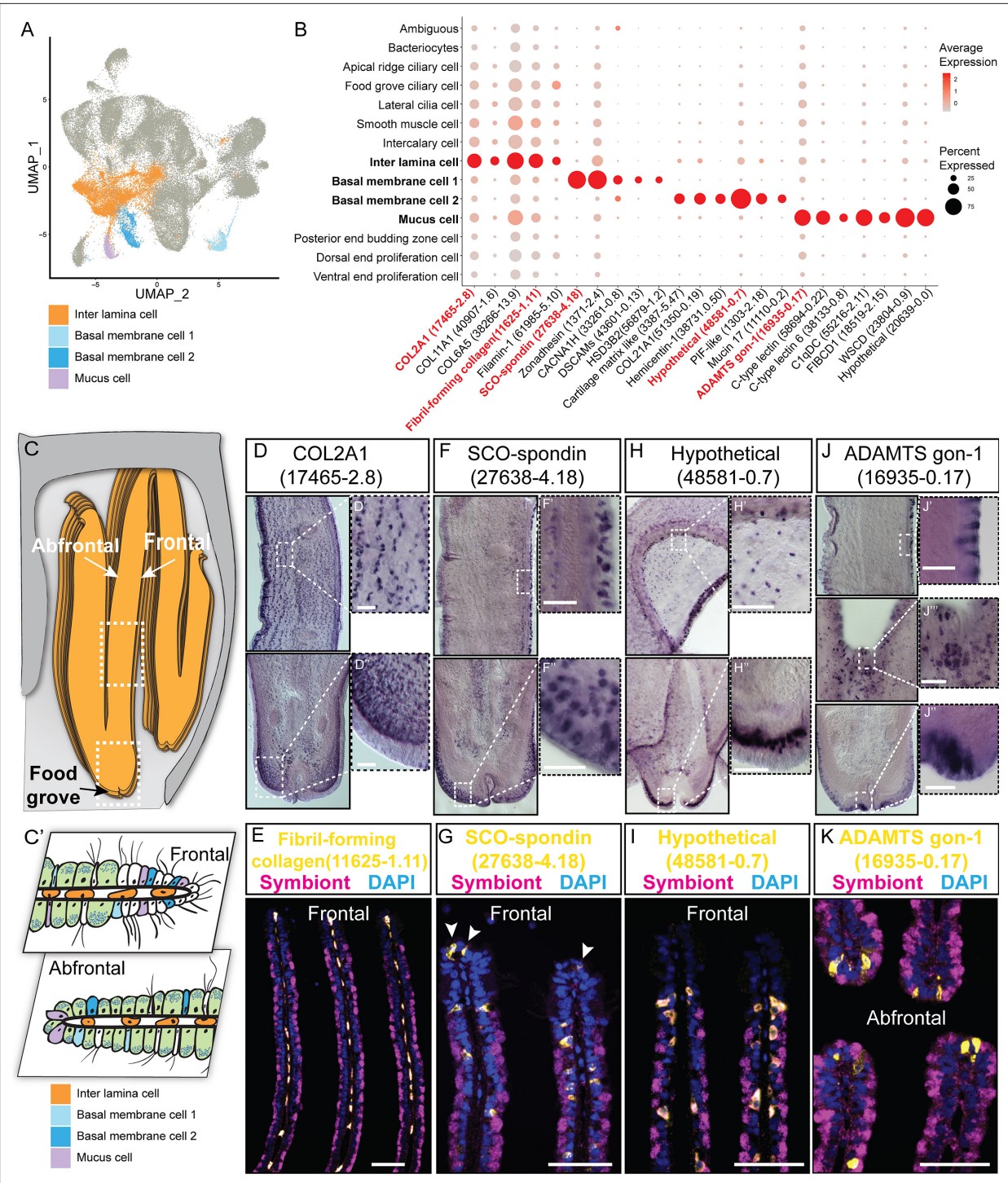

**Figure 2.** Supportive cell populations of *G. platifrons* gill. (**A**) Uniform Manifold Approximation and Projection (UMAP) representation of the four supportive cell populations. (**B**) Expression profiles of the cell markers that are specific or enriched in the supportive cell populations. The sizes of the circles represent the percentages of cells in those clusters that expressed a specific gene. Genes shown in red were validated by whole-mount in situ hybridisation (WISH) or double fluorescent in situ hybridisation (FISH). (**C, C'**) Schematics demonstrating the overall structural (**C**) and supportive cell distribution (**C'**). (**D, F, H, J**) WISH characterisation of the selected representative cell population markers. (**E, G, I, K**) Double FISH characterisation of the selected representative cell population markers. The white arrowheads in (**G**) indicate the BMC1 cells locates at the outer rim of gill slice. Scale bar: 50 μm.

The online version of this article includes the following figure supplement(s) for figure 2:

**Figure supplement 1.** Control hybridisation of the supportive cell markers.

We have also identified two types of BMC populations which have not been recognised before. The BMC1 (*Figure 2F and G*) expressed genes encoding extracellular matrix and adhesive proteins (*Figure 2B* and *Supplementary file 1b*), such as SCO-spondin (Bpl_scaf_27638-4.18), zonadhesin (Bpl_scaf_1371-2.4), cartilage matrix proteins (Bpl_scaf_16371-0.14 and Bpl_scaf_3387-5.47), and collagen alpha-1(XXI) chain protein (Bpl_scaf_51350-0.19). BMC2 (*Figure 2H and I*) was distinctive from BMC1 in the context of expression (*Figure 2A*) and marker genes (*Figure 2B*). Genes encoding hemicentins (Bpl_scaf_38731-0.50 and Bpl_scaf_7293-0.14), which are known to stabilise the cells' contact with the basal membrane (*Welcker et al., 2021*), were highly expressed in BMC2. Thus, these two cell types are likely to help build the basal lamina and stabilise the epithelial-derived cells, such as bacteriocytes and intercalary cells, on the surface of the basal membrane.

Mucus cells are specialised secretory cells with intracellular mucus vacuoles (*Fiala-Médioni et al., 1986*). In shallow-water filter-feeding bivalves, mucus cells secret mucus, which cooperates with the ciliary ventilation system to capture, process, and transport food particles to their mouths, known as filter feeding mechanism (*Dufour and Beninger, 2001*; *Dufour, 2005*; *Gómez-Mendikute et al., 2005*; *Beninger and Dufour, 1996*). Deep-sea mussels may not necessarily retain this function since they do not normally acquire food resources such as phytoplankton and planktonic bacteria but obtain most of the nutrients through their chemosynthetic symbiont. Thus, it was hypothesised that mucus cells are involved in other biological functions such as immune responses to pathogens (*Wang et al., 2021*). Herein, genes encoding proteins with microbe-binding functionalities were enriched in mucus cells (*Figure 2B* and *Supplementary file 1b*), such as C-type lectins (Bpl_scaf_58694-0.22 and Bpl_scaf_38133-0.8), C1q domain-containing protein (Bpl_scaf_55216-2.11), and fibrinogen C domain-containing protein (Bpl_scaf_18519-2.15) (*Gerdol et al., 2011*; *Wang et al., 2018*). The expression of similar immune genes was upregulated in a shallow-water mussel *Mytilus galloprovincialis* when challenged by a pathogenic bacteria (*Saco et al., 2020*). Our WISH and double FISH analyses showed that mucus cells were embedded within the outer rim cilia and scattered on the gill lamella alongside the bacteriocytes (*Figure 2J and K*). These data collectively suggest that mucus cells may help mussel maintain the immune homeostasis of gill. Interestingly, WISH and double FISH analyses showed that mucus cells were also distributed alongside the gill lamella's food groove and the inner edge, where the density of bacteriocytes is low (*Figure 2J*). Because the food groove is the main entry of food practical to the labial palps and mouth (*Richoux and Thompson, 2001*), this distribution pattern implies that mucus cells may be also involved in capturing planktonic bacteria and sending them to the mouth.

## Ciliary and smooth muscle cells

A remarkable feature of bathymodioline mussel's gill is its ciliary ventilation system, which constantly agitates the water and provides the symbiont with the necessary gas (*Riisgård et al., 2011*). We identified four types of ciliary cells (*Figure 3A and B*): apical ridge ciliary cells (ARCCs), food groove ciliary cells (FGCCs), lateral ciliary cells (LCCs), and intercalary cells (ICs), as well as a newly identified type of smooth muscle cells (SMCs). All ciliary cells were marked by canonical cilium genes, such as genes encoding flagellar proteins, ciliary motor kinesin proteins, ciliary dynein proteins, and ciliary microtubules and were clearly distinguishable by specifically expressed genes (*Supplementary file 1c* and *Figure 3—figure supplement 1*). ARCCs were characterised by the expression of tubulin alpha-1A chain Bpl_3489-0.37 (*Figure 3C*) and Bpl_scaf_20631-1.16, which encodes homeobox Dlx6a-like protein (*Figure 3D*) which is a marker of the apical ectodermal ridge (*Heude et al., 2014*). FGCCs, which could be ladled by expression of marker gene Bpl_scaf_5544-0.0 (*Figure 3E*), expressed genes encoding primary cilia development regulator Tubby-related protein 3 (*Han et al., 2019*) (Bpl_scaf_24834-2.3) and primary ciliary cell structural protein TOG array regulator of axonemal microtubules protein 1 (*Louka et al., 2018*; *Das et al., 2015*) (Bpl_scaf_55620-1.3), suggesting the FGCCs are the sensory ciliary cells that gather information from the surrounding environment. Both ARCCs and FGCCs were located around the ventral tip of the gill filament (*Figure 3F*). LCCs were distributed as two parallel rows along the gill lamella's outer rim and ciliary disks' outer rim (*Figure 3G and H*) and had highly expressed genes involved in cilium structure, cilium movement, and ATP synthases (*Figure 3B* and *Supplementary file 1c*), indicating that LCCs may have a strong ability to beat their cilia. Interestingly, we identified a group of previously unreported SMCs co-localised with the LCCs as showed by WISH (*Figure 3I*, *Figure 3—figure supplement 2*). SMCs strongly expressed several

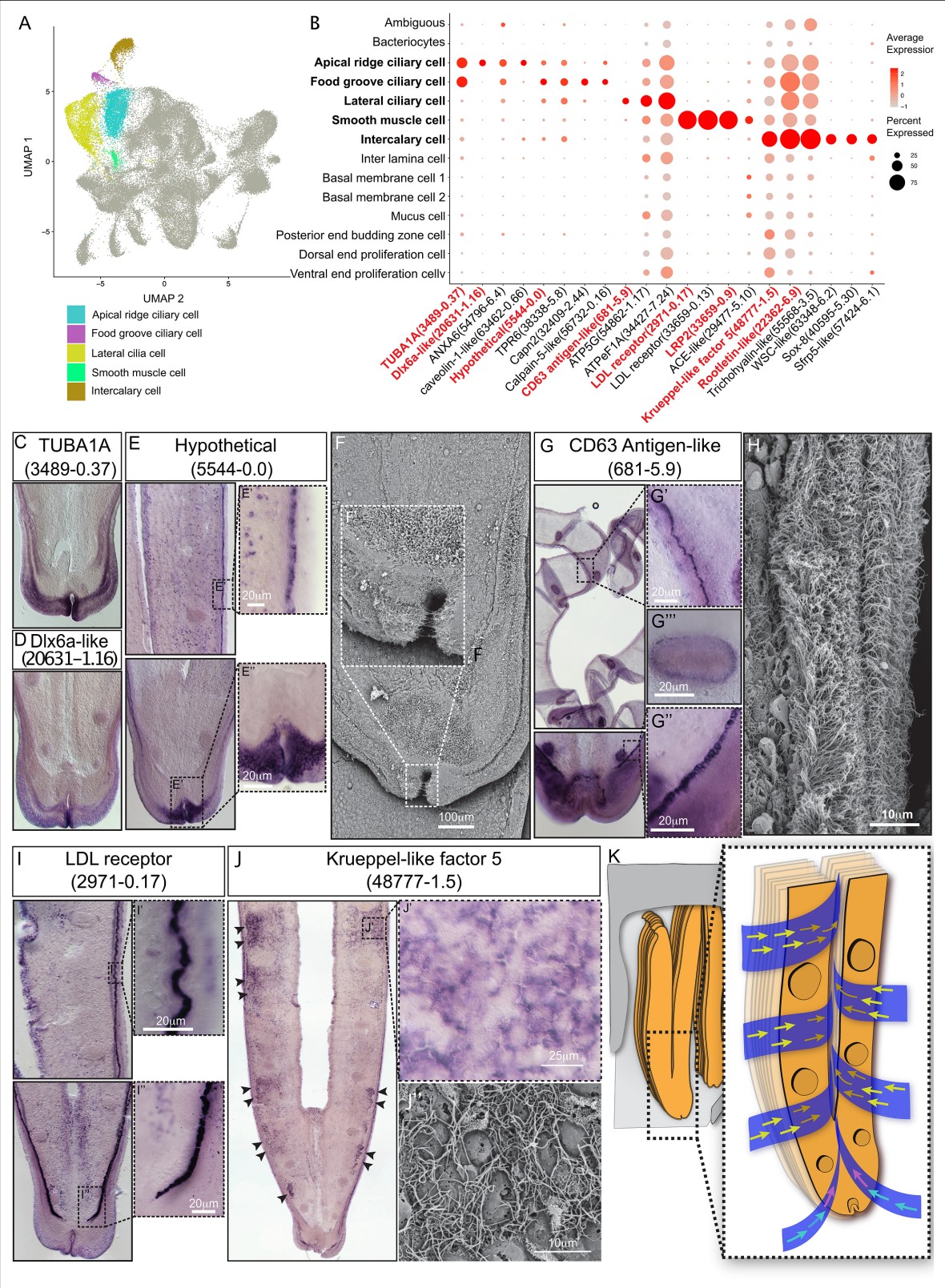

**Figure 3.** Ciliary cell populations of *G. platifrons* gill. (**A**) Uniform Manifold Approximation and Projection (UMAP) representation of the four ciliary cell populations and potential smooth muscle cell population. (**B**) Expression profiles of the cell markers that are specific or enriched in the ciliary cell populations. The sizes of the circles represent the percentages of cells in those clusters that expressed a specific gene. The genes shown in red were validated by whole-mount in situ hybridisation (WISH) or double fluorescent in situ hybridisation (FISH). (**C–E, G, I–J**) WISH characterisation of the

*Figure 3 continued on next page*

*Figure 3 continued*

selected representative cell population markers. (**F, H**) Scanning electron microscopy (SEM) analysis of the ciliary cells of *G. platifrons* gill. (**K**) Schematic of the water flow agitated by different ciliary cell types. The colour of the arrowheads corresponds to water flow potentially influenced by specific types of cilia, as indicated by their colour code in (**A**).

The online version of this article includes the following figure supplement(s) for figure 3:

**Figure supplement 1.** Heat map of the shared marker genes expressed in all the ciliary cells.

**Figure supplement 2.** Whole-mount in situ hybridisation (WISH) characterisation of the gene Bpl_scaf_48274-0.3, LRR2.

**Figure supplement 3.** Whole-mount in situ hybridisation (WISH) characterisation of gene Bpl_scaf_22362-6.9, Rootletin-like.

**Figure supplement 4.** Control hybridisation of the ciliary cell markers.

low-density lipoprotein receptors (LDL receptor) and LDLR-associated proteins (*Llorente-Cortés et al., 2000*; *Figure 3B*). In addition, these cells also expressed the angiotensin-converting enzyme-like protein (*Chen et al., 2016*) (Bpl_scaf-29477-5.10) and the 'molecular spring' titin-like protein (*Linke and Grützner, 2008*) (Bpl_scaf_56354-7.8) (*Supplementary file 1c*). The expression of these genes could be commonly found in human vascular SMCs (*St Paul et al., 2020*; *Ytrehus et al., 2021*; *Keller et al., 2000*). Collectively, we suspect that SMCs are involved in lateral cilium movement and the gill slice contraction.

ICs are the specialised ciliary cells surrounding the bacteriocytes (*Figure 3B and J*). WISH analysis showed that the expressions of IC markers had apparent spatial variations (*Figure 3J*). The expression of ICs marker rootletin (*Chen et al., 2015b*; *Styczynska-Soczka and Jarman, 2015*; *Mohan et al., 2013*) (Bpl_scarf_22362-6.9) was considerably higher in the ICs close to the frontal edge of the gill filament, and the expression gradually decreased along with the direction of inter lamina water flow (*Figure 3K*, *Figure 3—figure supplement 3*), implying that ICs ventilate the water flow and the mucus through the gill filaments. Furthermore, compared with the other three types of ciliary cells, the ICs expressed several genes encoding transcription factors involved in determining cell fate (*Figure 3B* and *Supplementary file 1c*), such as transcription factors Sox 8 (Bpl_scarf_40595-5.30) and Wnt pathway cell polarity regulator secreted frizzled-related protein 5 (Bpl_scaf_57424-6.1) (*Jones and Jomary, 2002*), suggesting that the ICs might also play regulatory roles (*Gillis et al., 2008*; *Zhang et al., 2017*; *Phochanukul and Russell, 2010*).

## Proliferation cells

It has long been known that the bathymodioline mussel gill has three types of proliferation cells that are conserved throughout all filibranchia bivalves: the budding zone at the posterior end of the gill where new gill filaments are continuously formed, and the dorsal and ventral ends of each gill filaments (*Cannuel et al., 2009*; *Leibson and Movchan, 1975*). These 'cambial-zone'-like cell populations could continuously proliferate throughout the whole life span of the mussel (*Leibson and Movchan, 1975*). Our snRNA-seq data recognised three types of proliferation cells, which is consistent with previous findings (*Figure 4A and B* and *Supplementary file 1d*). The gill posterior end budding zone cells (PEBZCs) are located on the first few freshly proliferated filaments of the posterior tip of gill (*Figure 4C and D*). The PBEZCs marker (Bpl_scaf_61993-0.4) gradually disappeared at around the 11th–12th row of the gill filament (*Figure 4D*), suggesting the maturation of the gill filaments, which is similar to the developmental pattern reported in another deep-sea mussel *Bathymodiolus azoricus* (*Wentrup et al., 2014*). The dorsal end proliferation cells (DEPCs), which expressed the hallmarks genes of muscular tissue (*Figure 4B*), as well as cell proliferation and differentiation regulators, were the proliferation cells in connective tissue at the dorsal end of the gill slice (*Figure 4A and E*). The ventral end proliferation cells (VEPCs) were two symmetrical triangle-shaped cell clusters of small symbiont-free cells (*Figure 4F*). In VEPCs, genes encoding ribosomal proteins, chromatin proteins, RNA and DNA binding proteins, and cell proliferation markers were all upregulated (*Figure 4B* and *Supplementary file 1d*), indicating that VEPCs are meristem-like cells (*Wentrup et al., 2014*; *Mohieldin et al., 2020*).

It has been hypothesised that new proliferation cells will be colonised by symbionts, serving as the vital mechanism of bacteriocyte recruitment (*Wentrup et al., 2014*). We then determined which proliferation cell type gives rise to the mature cells, especially the bacteriocytes, which has little available information regarding their precursor and transmission mode in bathymodioline mussels (*Wentrup et al., 2014*; *Neumann and Kappes, 2003*). We performed Slingshot analysis, which uses a

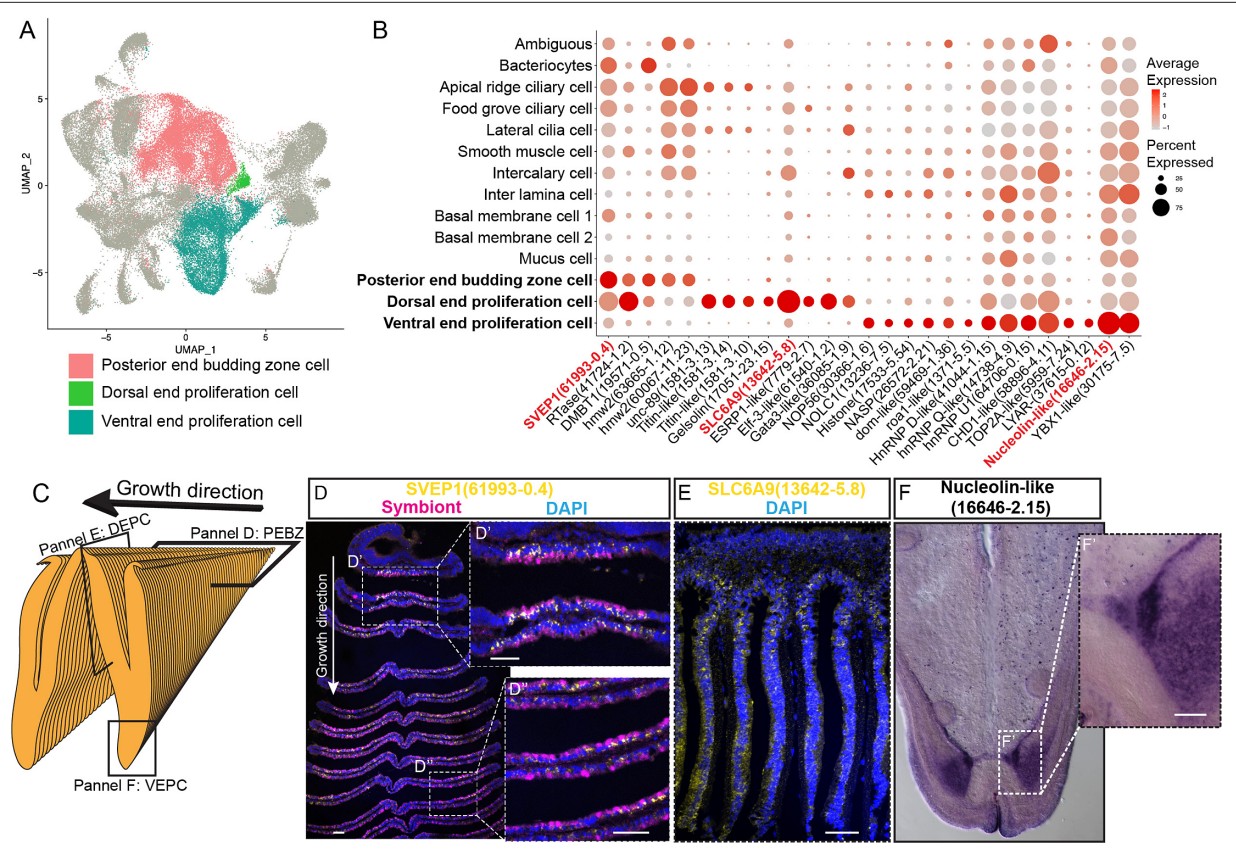

**Figure 4.** Proliferation cell populations of *G. platifrons* gill. (**A**) Uniform Manifold Approximation and Projection (UMAP) representation of the three proliferation cell populations. (**B**) Expression profiles of the cell markers that are specific or enriched in the supportive cell populations. The sizes of the circles represent the percentages of cells in those clusters that expressed a specific gene. Genes shown in red were validated by whole-mount in situ hybridisation (WISH) or double fluorescent in situ hybridisation (FISH). (**C**) Schematic analyses of the spatial position of the three Proliferation cell populations. (**D, E, F**) FISH and WISH characterisation of the selected population markers. The marker genes confirmed by ISH or WISH in the current study are indicated in red. Scale bar: 50 μm.

The online version of this article includes the following figure supplement(s) for figure 4:

**Figure supplement 1.** Slingshot trajectory of selected cell types (a dot represents a cell) identified by Slingshot and PHATE.

**Figure supplement 2.** Control hybridisation of the proliferation cell markers.

cluster-based minimum spanning tree (MST) and a smoothed principal curve to determine the developmental path of cell clusters. The result shows that the PEBZCs might be the origin of all gill epithelial cells, including the other two proliferation cells (VEPC and DEPC) and bacteriocytes (*Figure 4—figure supplement 1*). The sole exception was BMC2, which may be derived from VEPC rather than PEBZC. This result is consistent with previous studies which suggested new gill filaments of the filibranch mussels are formed in the gill's posterior budding zones (*Wentrup et al., 2014*). The colonisation by the symbiont might play a crucial role in determining the fate of the bacteriocytes. Noticeably, in *G. platifrons*, only the pillar-shaped first row of gill filament, comprised of small meristem-like cells, was symbiont-free (*Figure 4D and D'*), whereas all the other gill filaments were colonised by symbionts. This pattern of symbiosis establishment is different from that of *B. azoricus*, in which PEBZCs are symbiont free and are gradually colonised by symbionts released from the bacteriocytes on the adjacent mature gill filaments after maturation (*Wentrup et al., 2014*). On mature gill filaments, the DEPCs and VEPCs are seemingly the sources of new cells that sustain the growth of the gill filament from both dorsal and ventral directions, respectively (*Cannuel et al., 2009*). Interestingly, comparable active ventral and dorsal end proliferation zones have also been identified in the symbiotic mussel *B. azoricus*, whereas they are absent in the shallow-water mussel *Mytilus edulis* (*Piquet et al., 2020*).

This contrast further suggests the potential involvement of DEPCs and VEPCs in the establishment of symbiosis.

## Bacteriocytes and host–symbiont interaction

We conducted the whole-mount FISH using a bacterial 16S rRNA probe for symbiont to determine the spatial distribution of bacteriocytes, and their positions relative to the other cell types on the gill filament (*Figure 5A and B*). Bacteriocytes covered the majority of the surface of the gill filament, except the ventral tip, the ciliary disk, and the frontal edge (lateral ciliary). The bacteriocytes were surrounded by intercalary cells with microvilli and cilia on the surface (*Figure 5C*).

The gene expression profile of bacteriocytes aligned well with the ultrastructural analysis, which suggested that the bacteriocytes have structural, metabolic, and regulatory adaptions to cope with the symbiont (*Figure 5D–I*). It has been hypothesised that the bacteriocytes extract nutrients from the symbiont through two pathways: lysing the symbiont through the lysosome (the 'farming' pathway) or directly utilising the nutrient produced by the symbiont (the 'milking' pathway) (*Streams et al., 1997*; *Ponnudurai et al., 2017*). Our TEM observations clearly detected intracellular vacuole and lysosome system of the bacteriocyte that harbour, transport, and digest the symbionts (*Figure 5D*), which is consistent with previous studies (*Wang et al., 2021*; *Barry et al., 2002*). The snRNA data showed bacteriocytes expressed cellular membrane synthesis enzyme (phosphoethanolamine N-methyltransferase, Bpl_scaf_15282-3.10) and a series of lysosomal proteins such as lysosomal proteases (cathepsins; Bpl_scaf_61711-5.12, Bpl_scaf_46838-6.40, and Bpl_scaf_59648-4.5; lysosomal alpha-glucosidase-like), protease regulators 56 (TMPRSS15 proteins; Bpl_scaf_52188-1.15 and Bpl_scaf_15410-0.9), and lysosomal traffic regulator proteins (rabenosyn-5, Bpl_scaf_54816-0.3 and Bpl_scaf_52809-1.6, lysosomal-trafficking regulator-like isoform X1). Among the proteases, cathepsins were thought to be evolutionary conserved molecular tools that host utilised to control the residence of their symbiont microbes (*Renoz et al., 2015*). They were also highly expressed in symbiotic tissue of other deep-sea chemosynthetic animals, such as vesicomyid clams and vestimentiferan tubeworms (*Guan et al., 2022*; *Li et al., 2019*; *Sun et al., 2021*). Bacteriocytes also expressed genes encoding cellular vesicle transports (kinesins, Bpl_scaf_14819-0.11, Bpl_scaf_54265-1.13, and Bpl_scaf_4784-1.40) (*Jones and Jomary, 2002*), potential amino acid transporter (*Cannuel et al., 2009*) (Bpl_scaf_36159-5.5), and genes involved in intracellular vesicle transport, such as the FYVE and coiled-coil domain-containing protein 1 (*Leibson and Movchan, 1975*) (Bpl_scaf_33726-5.7). In addition to form and mobilise early endosomes, the protein products of these genes could transport symbiont-secreted nutrient vesicles to the host (*Figure 5F* and *Supplementary file 1e*), supporting the 'milking' pathway.

The symbiont of *G. platifrons* belongs to type I methanotrophy, of which the core metabolic function is linked with the development of intracytoplasmic membranes leading to a high lipid/biomass content (*Takishita et al., 2017*; *Demidenko et al., 2016*). Recent lipid biomarker analyses showed that the gill of *G. platifrons* contains a high amount of bacterial lipids, which are directly utilised by the host to synthesise most of its lipid contents (*Guan et al., 2022*). Downstream of the bacteriocyte's metabolic cascade, genes encoding proteins that may be involved in fatty acid/lipid metabolism, such as perilipin2 (Bpl_scaf_27158-3.8), which is the critical protein to form intracellular lipid droplets (*Brasaemle et al., 1997*), and a variety of fatty acid metabolism enzymes (acetyl carboxylase 2, Bpl_scaf_55250-5.11 [*Cheng et al., 2007*]; fatty acid desaturase 1-like isoform X1, Bpl_scaf_35916-0.6 [*Monroig and Kabeya, 2018*; *Kabeya et al., 2020*]; long-chain fatty acid-ligase ACSBG2-like isoform X1, Bpl_scaf_28862-1.5 [*Soupene and Kuypers, 2008*; *Alves-Bezerra et al., 2016*]), were upregulated, suggesting that the fatty acid could be a major form of nutrients passing from the symbiont to the host mussel.

Additionally, bacteriocytes expressed several solute carriers, including sodium/ascorbate cotransporter (*Figure 5I*, solute carrier family 23 members 1-like, Bpl_scaf_32311-1.19), sodium/potassium/calcium exchanger (sodium potassium calcium exchanger 3-like, Bpl_scaf_14503-1.28), sodium/chloride ion cotransporter (solute carrier family 12 member 3-like isoform X1, Bpl_scaf_44604-3.7), ferrous iron transporter (solute carrier family 40 member 1-like, Bpl_scaf_63447-0.12), and zinc transporter (zinc transporter ZIP14-like, Bpl_scaf_44428-5.3). The solute carriers are a large family of ATP-dependent transporters that shuttle a variety of small molecules across the cellular membrane (*Höglund et al., 2011*). In several model symbiotic systems, solute carriers play a vital role in host–symbiont interaction by either providing the symbiont with substrates (*Mohamed et al., 2020*; *Bertucci et al.,*

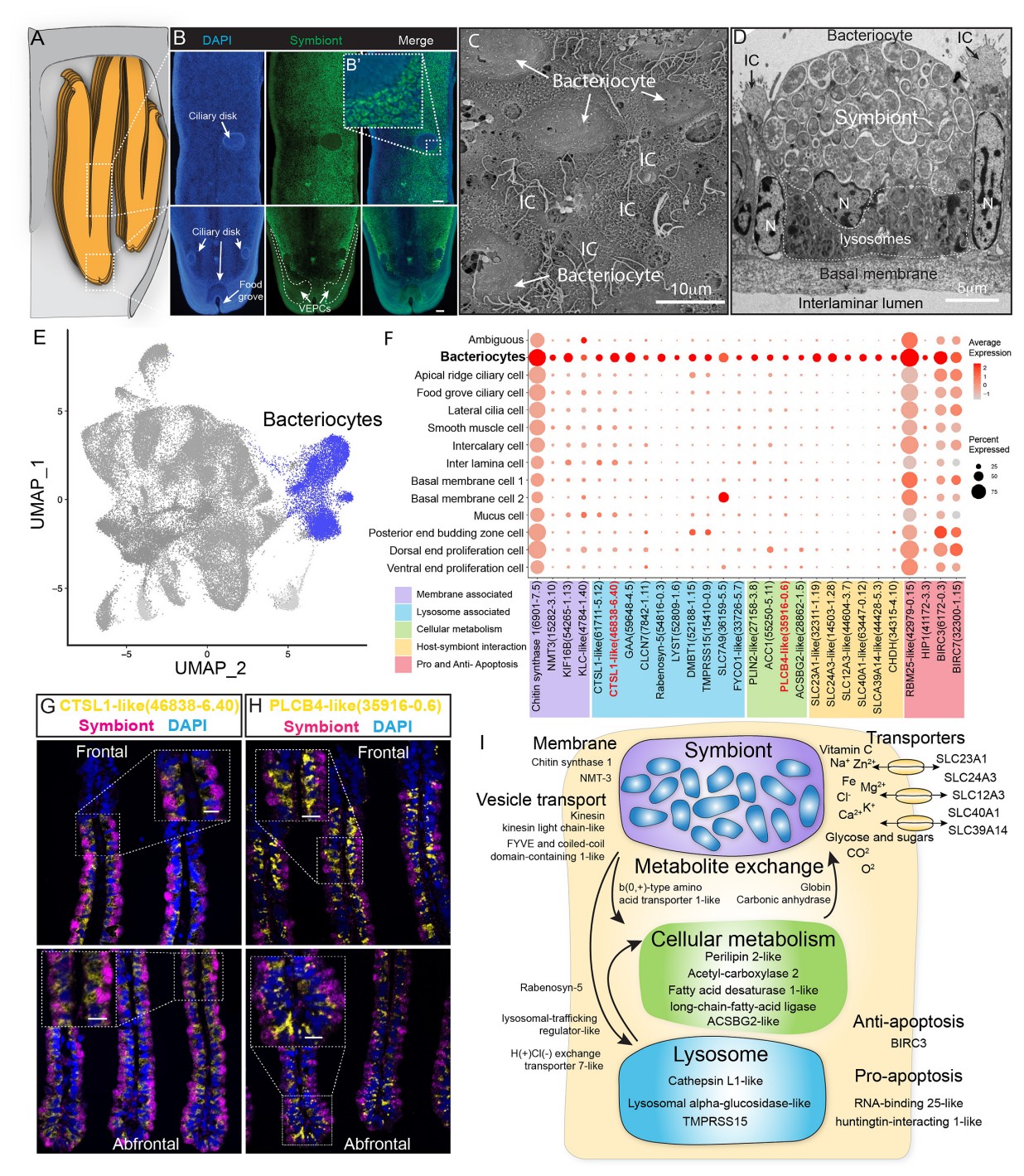

**Figure 5.** Characterisation of the bacteriocytes of *G. platifrons*. (**A**) Schematic of the overall structure of *G. platifrons* gill filaments. (**B**) Whole-mount fluorescent in situ hybridisation (FISH) analyses of the overall distribution of bacteriocytes on the *G. platifrons* gill filament. (**C**) Scanning electron microscopy (SEM) analysis of the bacteriocytes. (**D**) Transmission electron microscopy (TEM) analysis of a bacteriocyte. (**E**) Uniform Manifold Approximation and Projection (UMAP) representation of *G. platifrons* bacteriocytes. (**F**) Expression profiles of the cell markers that are specific or enriched in the bacteriocytes. The sizes of the circles represent the percentages of cells in those clusters that expressed a specific gene. (**G, H**) Double FISH validated the genes shown in red. (**I**) Schematic of the host–symbiont interaction based on the single-cell transcriptome of *G. platifrons* bacteriocytes. The marker genes confirmed by ISH in the current study are indicated in red. Scale bar in panels 25 μm.

The online version of this article includes the following figure supplement(s) for figure 5:

**Figure supplement 1.** Control hybridisation of the bacteriocyte markers.

2015) or transporting symbiont-produced nutrients to the host (*Hamada et al., 2018*; *Duncan et al., 2016*; *Feng et al., 2019*). Previous studies demonstrated that solute carrier genes were expanded in deep-sea chemosynthetic animals' genome (*Ip et al., 2021*) and highly expressed in symbiotic organs (*Hongo et al., 2016*), including bathymodioline mussel's gill (*Zheng et al., 2017*). In bathymodioline mussels, previous bulk RNA-seq studies detected upregulated expression of a large variety of solute carriers in the gill, which is consistent with the present study, suggesting that solute carriers may play crucial roles in shuttling nutrients in and out of bacteriocytes and in maintaining the suitable intracellular micro-environment (such as the SLC23A1 and SLC39A14) for the symbiont (*Sotiriou et al., 2002*; *Aydemir et al., 2012*).

## Cell-type-specific response to environmental stresses

To examine cell-type-specific acclimatisation to environmental changes, the expression profiles of differentially expressed genes (DEGs) were compared between the three states (Fanmao, starved, and reconstituted animals) of samples collected in our in situ transplantation experiment (full lists of DEGs are shown in *Supplementary file 1f–r*). Notably, for each state three mussels were processed but pooled for nuclei extraction before snRNA sequencing (see 'Materials and methods' for detail). We tested our hypothesis here that at the starvation site a relatively low concentration of methane shall upset symbiont metabolism and thus substantially affect symbiont-hosting bacteriocytes by assessing the transcriptional changes per cell type. We calculated the centroid coordinates for each cell type in each state on the two-dimensional Uniform Manifold Approximation and Projection (UMAP) plot (*Figure 6A*). Then, for each cell type, we determined the Euclidean distance between the centroid coordinates of each pair of states (*Supplementary file 1s*). The impact of starvation was variable across cell types, as reflected by the cross-state distances (*Figure 6B*, green bar). Starvation resulted in the most significant transcriptional changes in bacteriocytes reflected by a large Fanmao-vs.-starvation distance (2.3), followed by VEPC and inter lamina cells (2.1 and 1.4, respectively). On the other hand, starvation had little impact on the transcriptions of ciliary cells and most supportive cells such as the food grove ciliary cell, BMC2, and mucus cells (distances <0.5). *Figure 6B* also shows that after reconstitution (moving the mussels back to the methane-rich site Fanmao for 3 days), the expressional profile of bacteriocytes rapidly changed back, reflected by a large starvation-vs.-reconstitution distance and much smaller Fanmao-vs.-reconstitution distance (2.3 vs. 0.8; *Supplementary file 1s*). This result coincided with and was supported by our pseudo-time analysis for bacteriocytes (*Figure 6C*), showing that bacteriocytes in the reconstitution are in intermediate and transitional states between Fanmao and starvation.

For the bacteriocytes population, we conducted cell trajectory pseudotiming and detailed DEG analysis to interpret the mechanism of host–symbiont interactions. It is important to acknowledge that our sampling strategy might have limitations as more closely spaced time points could enhance the confidence of trajectory reconstruction. The branched heat map showed both up- and down-regulated genes during starvation and reconstitution compared with the Fanmao state (*Figure 6D*, *Figure 6—figure supplement 1*). The Kyoto Encyclopedia of Genes and Genomes (KEGG) pathway enrichment analysis of the bacteriocytes' DEGs provides an overall view of the pathways enriched in each environmental state (*Figure 6—figure supplement 2A–C*). The genes encoding ribosomal proteins are highly expressed in the bacteriocytes under the methane-rich 'Fanmao' state (*Figure 6—figure supplement 2A*), suggesting an active protein synthesis and cellular metabolism (*Turi et al., 2019*). Moreover, organic ion transporters (A BCB P-glycol, Bpl_scaf_18613-16.10; canalicular multispecific organic anion transporter 2, Bpl_scaf_52110-4.43; sodium- and chloride-dependent glycine transporter 1-like, Bpl_scaf_13642-5.8), which may be involved in transporting symbiont-produced nutrients, are also highly expressed (*Figure 6—figure supplement 3*).

In starved *G. platifrons*, negative regulators of cell proliferation, such as autocrine proliferation repressors and receptor-type tyrosine phosphatase beta, were upregulated in bacteriocytes, suggesting repression of cell growth. We also observed the enrichment of genes in the apoptosis pathway (*Figure 6—figure supplement 2B*), which is attributed to the upregulation of caspases (Bpl_scaf_25165-3.8 and Bpl_scaf_24225-0.12) and cathepsins (Bpl_scaf_64706-0.16, Bpl_scaf_61711-5.12) which can trigger caspases-dependent cell death, suggesting cellular stress condition. Correspondingly, a gene encoding baculoviral IAP repeat-containing protein (Bpl_scaf_6172-0.3), which can bind caspase and inhibit apoptosis, was upregulated in bacteriocytes (*Figure 6E*). In *Drosophila*, the

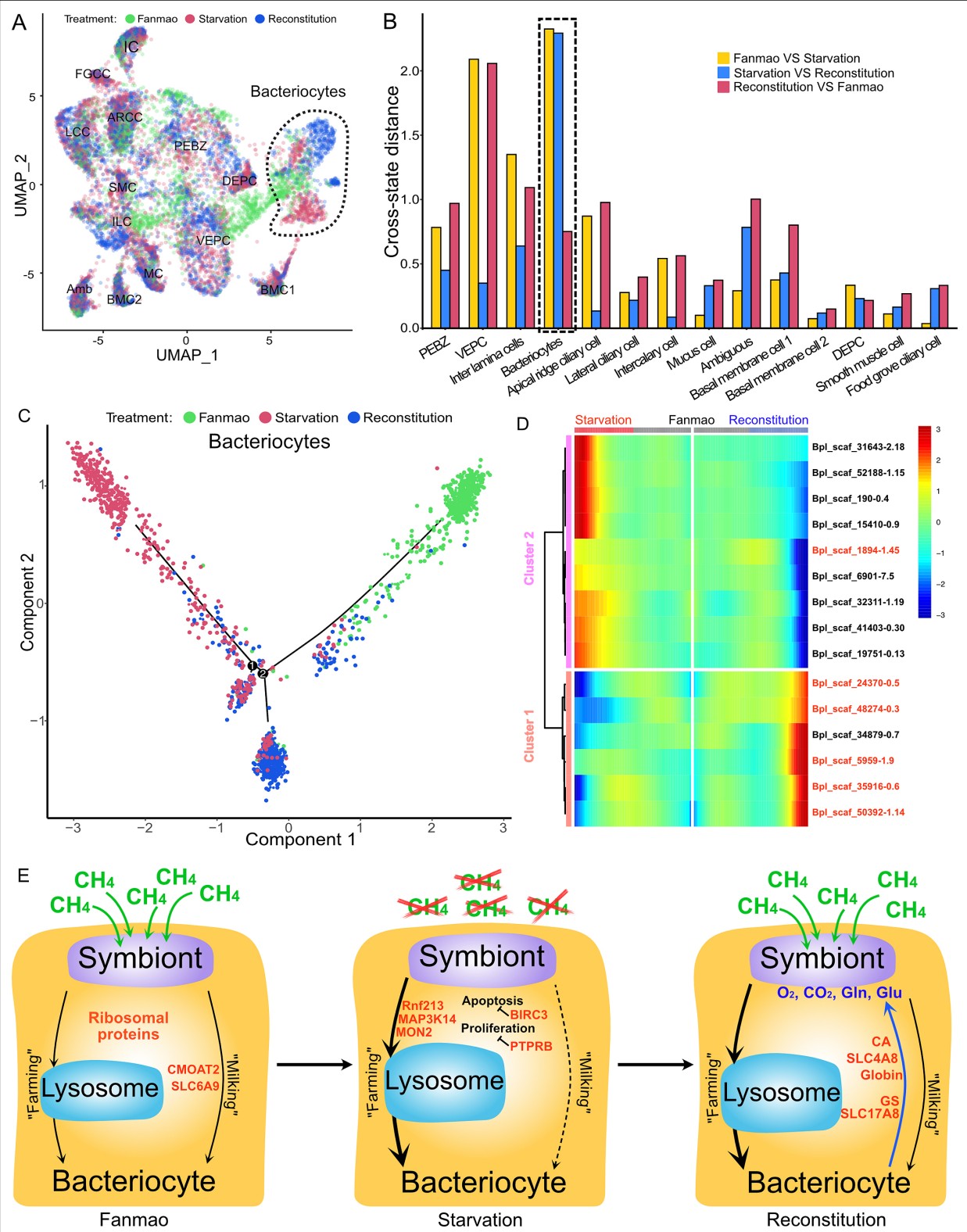

**Figure 6.** Analysis of cell population-specific differentially expressed genes (DEGs). (**A**) Uniform Manifold Approximation and Projection (UMAP) representation of the impact of deep-sea in situ transplant treatments on the gene expression pattern of each cell population. The cells from different treatments are labelled with different colours. The dashed line encircled bacteriocyte populations have a considerably altered expression profile. (**B**) Histogram of cross-state distances between the centroids of the Fanmao, starvation, and reconstitution groups per cell type on UMAP. The black dashed lines indicate the bacteriocyte populations whose expression profile was remarkably altered. (**C**) Visualisation of bacteriocytes onto the

*Figure 6 continued on next page*

*Figure 6 continued*

pseudotime map using monocle. The black lines indicate the main path of the pseudotime ordering of the cells. (**D**) Bifurcation of selected gene expression along two branches in response to environmental perturbation. Genes are clustered hierarchically into two groups, illustrating up- (cluster 1) and down- (cluster2) regulated genes in the starvation state compared with Fanmao. Genes in red colour are discussed in the section 'Cell-type-specific response to environmental stresses'. The heat map showing the gene expression profiles of all bacteriocytes' DEGs is shown in *Figure 6—figure supplement 1*. (**E**) Proposed model for the molecular mechanisms of host–symbiont interactions in response to environmental changes.

The online version of this article includes the following figure supplement(s) for figure 6:

**Figure supplement 1.** Heat map showing the gene expression profiles of all bacteriocytes' differentially expressed genes (DEGs).

**Figure supplement 2.** Kyoto Encyclopedia of Genes and Genomes (KEGG) enrichment analysis of the bacteriocytes' differentially expressed genes (DEGs).

**Figure supplement 3.** Gene expression-level analysis of selected genes.

**Figure supplement 4.** An RAxML phylogenetic tree estimated using E3 ubiquitin ligase RNF213.

baculoviral IAP proteins are important in animal's response to cellular stress and promote cell survival (*Hay, 2000*; *Dubrez-Daloz et al., 2008*). Similarly, the gene encoding MAP3K14 (Bpl_scaf_60908-4.14) and potential E3 ubiquitin ligase RNF213 (Bpl_scaf_25983-1.26, Bpl_scaf_1894-1.45, *Figure 6—figure supplement 4*), which were apoptosis suppressor or regulator, were upregulated (*Pflug and Sitcheran, 2020*). These results suggest that starved bacteriocytes were carrying out cell-type-specific adjustments to cope with stresses. The E3 ubiquitin ligases could also work as intracellular immune sensors of bacterial lipopolysaccharides (*Otten et al., 2021*). Thus, the encoded protein may be able to active downstream immunological toolkit to digest the symbiont population for nutrients.

As mentioned above, bacteriocytes obtain nutrients from endosymbionts. KEGG pathway analyses suggested that phagosome, lysosome-related pathways were upregulated in the starvation state, indicating that bacteriocytes were actively digesting the endosymbionts in the starved *G. platifrons*. Interestingly, protein synthesis activity was more activated in inter lamina cells and VEPCs as supported by the upregulation of genes encoding ribosomal proteins in both cell populations (*Supplementary file 1f and g*). This was contrary to the situation in bacteriocytes and may be a consequence of the high activity of 'farming' in bacteriocytes.

We anticipated that after moving the starved *G. platifrons* back to Fanmao site, bacteriocytes and their endosymbionts would 'reconstitute', leading to the partly restored 'farming' and 'milking' pathways. This was confirmed by higher expression of fatty acid metabolic genes in the mussels in the reconstitution state than those in the starvation state, such as long-chain fatty acid ligase ACSBG2-like (Bpl_scaf_28862-1.5), elongation of very long-chain fatty acids 7-like (Bpl_scaf_5959-1.9), and fatty acid desaturase 1-like (Bpl_scaf_35916-0.6). These findings were also consistent with the result of KEGG analyses. The mitochondrial trifunctional enzyme (Bpl_scaf_42376-0.5) was also highly expressed, suggesting a high level of energy-producing activity. The KEGG analyses showed that glutamatergic synthase was enriched in the reconstitution state in comparison with those in both the Fanmao and starvation states. In the insect aphid–Buchnera endosymbiosis model, the host-produced glutamate could be transported and directly utilised by the symbiont (*Price et al., 2014*). Similar glutamate-based host–symbiont metabolic interaction mechanisms were also proposed in the deep-sea mussel *Bathymodiolus thermophilus* and *B. azoricus* (*Ponnudurai et al., 2017*; *Ponnudurai et al., 2020*). As the pseudotime analyses showed above (*Figure 6C*), reconstitution was the intermediate state between Fanmao and starvation. Metabolic and gene regulatory functions of bacteriocytes were still different from that in Fanmao. The KEGG analyses suggested regulatory (Rap1 signalling, retrograde endocannabinoid signalling, chemokine signalling, glucagon signalling, thyroid hormone synthesis, mRNA surveillance pathway, etc.) and metabolic pathways were enriched in the reconstitution state (fatty acid metabolism, salivary secretion, aldosterone synthesis and section, insulin secretion and pancreatic section) (*Figure 6—figure supplement 2C*). For example, we detected upregulation of the genes encoding carbonic anhydrases (Bpl_scaf_33596-7.3 and Bpl_scaf_48274-0.3), electroneutral sodium bicarbonate exchanger 1 (Bpl_scaf_61230-5.11), and globin-like proteins (Bpl_scaf_50392-1.14 and Bpl_scaf_24370-0.5), which could provide the symbiont with carbon dioxide and oxygen necessary for the symbiont's chemosynthetic metabolism (*Ip et al., 2021*; *Tashian, 1989*).

## Summary and outlook

Using the deep-sea mussel *G. platifrons* as a model organism, we demonstrated the power of integrating snRNA-seq and WISH data in unravelling the mechanism behind animal–microbe symbiosis. The robustness of this strategy showed stable and highly distinguishable expression patterns of each cell type regardless of the different environmental states. We successfully profiled the specific roles of different types of cells, including the previously unknown cell types, in maintaining the structure and function of the gill. The supportive cells that are located in between (inter lamina cells) and on the basal membrane's surface (BMC1 and BMC2) helped maintain the anatomical structure of the basal membrane. Ciliary cells and SMCs involved in gill slice contraction and cilium beat allow the gathering of material and information from the surrounding environment. Proliferation cells (PBEZCs, DEPCs, and VEPCs) gave rise to new cells, including bacteriocytes which obtained nutrients from endosymbionts using intracellular vacuoles and lysosomes.

The snRNA analyses also revealed that different cell types collaborated to support bacteriocytes' functionality. The PBEZCs gave rise to new bacteriocytes that allowed symbiont colonisation. Bacteriocytes attached to the basal membrane and were stabilised by supportive cells. Mucus cells co-localise with bacteriocytes and help maintain immune homeostasis. The beating ciliary cells controlled the water flow, providing bacteriocytes with necessary inorganic substances from the environment. This new information on cell–cell interaction certainly advanced our overall understanding of how endosymbiotic microbes and host cells communicate and collaborate, which cannot be easily achieved through other methods.

Moreover, the analysis of snRNA data from bacteriocytes has provided insight into the molecular mechanisms employed by the host to maintain and regulate the symbiosis. Notably, the bacteriocytes-enriched transcripts involved in harbouring, digesting symbionts, and transporting nutrients produced by the symbionts were identified and characterised. Our in situ transplant experiments by moving mussels between methane-rich and methane-limited sites also provided clues of cell-type-specific responses to environmental change. Under a methane-limited environment, the staved mussels more actively consumed endosymbionts through the 'farming' pathway. After being moved back to the methane-rich environment, mussels produced more glutamates which sustained the regrowth of symbionts. These preliminary findings showed that the deep-sea mussels were able to control their endosymbionts using a set of genes in response to environmental change. Due to the limitation of remotely operated vehicle (ROV) dives and sampling capacity, we had to have pooled samples of each state for nuclei extraction and sequencing. Thus, the cells per cluster could be considered as technical than biological replicates. Although this sampling process strategy has been broadly used in snRNA/scRNA sequencing (*Hicks et al., 2018*), we recognise the possible violation of assumptions in p-value calculation for DEGs between the three states.

Overall, the single-cell spatial and functional atlas developed in the present work will decipher some common principles of symbiosis and environmental adaption mechanisms of animals. The workflow developed in the present study could provide insightful references for researchers focusing on the mechanistic study of the biological adaptation of biologically and ecologically important non-model animals.

## Materials and methods

### Deep-sea in situ transplant experiment and sample collection

In situ transplant experiment was conducted at the 'F-site' cold seep during the *R/V 'Kexue'* 2020 South China Sea cold seep cruise. The overall design of the in situ transplant experiment and environmental states is shown in *Figure 1*. First, mussels in the methane-rich 'Fanmao' site (meaning 'prosperous' site, 22°06′55.425″N,119°17′08.287″E, depth 1117 m) were scoped into three nylon bags with approximately 10 mussels in each bag. Then, two bags of mussels were transplanted to the low-methane starvation site (*Figure 1*, 22°07′00″N, 119°17′07.02″E, depth 1147.42 m). After 11 days of transplantation, one bag of mussels in the starvation site was moved back to the 'Fanmao' site. On the 14th day of the transplant, three bags of mussels – one bag from the Fanmao site (designated as the 'Fanmao' sample), one bag from the starvation site (designated as the 'starvation' sample), and one bag of mussels which were first transplanted to the starvation site for 11 days and moved back to the Fanmao site for 3 days (designated as the 'reconstitution' sample) – were all retrieved by ROV Fanxian.

The mussels were kept in a hydraulic pressure-sealed biobox during the ascending of the ROV. The biobox is made of heat-insulated material, which will prohibit heat exchange with warm surface water. The samples were immediately processed once onboard *R/V Kexue*. For snRNA-seq, the posterior end tip of the mussel's gill was dissected (*Figure 1C*), snap-frozen with liquid nitrogen, and then stored at −80°C until use. For WISH, ISH, and FISH, the gills of the mussels were first fixed with 4% paraformaldehyde (PFA, prepared with autoclaved 0.22 μM membrane filtered in situ seawater) at 4°C overnight. Then, the gill was washed with ice-cold 1× PBS three times, dehydrated, and stored in 100% methanol at −20°C.

## Single-nucleus RNA-sequencing of *G. platifrons*

The gill nucleus was extracted using the Nuclei PURE prep nuclei isolation kit (Sigma-Aldrich). For each sample, posterior end tips from three individual mussels were randomly selected and pooled together. The posterior tips were homogenised in 10 mL of ice-cold lysis solution (Nuclei PURE lysis buffer with 0.1% Triton X100 and 1 mM DTT) on ice. The cell nuclei were then separated by sucrose gradient centrifugation according to the manufacturer's protocol. The cell nuclei pellets were washed by re-suspending in ice-cold DPBS–BSA solution (1× DPBS, 0.04% nuclease-free BSA, 0.01% RNase inhibitor; Takara). The nucleus was spun down by centrifugation at $500 \times g$ for 5 min at 4°C. This step was repeated to remove the contaminants from the cell plasma. Finally, the nucleus was re-suspended in the DPBS–BSA solution. The concentration of the cell nucleus was counted using Cell Countess II. Single-nucleus RNA-seq libraries were then constructed with the BD Rhapsody single-cell analysis system using the BD Rhapsody WTA Amplification Kit according to the manufacturer's protocol. The library was subjected to 150 bp paired-end sequencing using the Illumina HiSeq 4000 platform. The clean reads of three datasets were submitted to the National Centre for Biotechnology Information Sequence Read Archive database (Bioproject: PRJNA755857).

## Bioinformatics

The three raw datasets (Fanmao, starvation, and reconstitution) were processed individually following the BD single-cell genomics analysis setup user guide (Doc ID: 47383 Rev. 8.0) and BD single-cell genomics bioinformatics handbook (Doc ID: 54169 Rev. 7.0). This process involved the preparation of gene names, alignment of data to the genome, and generation of an expression matrix for each dataset. A *G. platifrons* genome reference v1.0 (available at the Dryad Digital Repository; http://dx.doi.org/10.5061/dryad.h9942) was utilised. The genome was indexed and constructed using STAR 2.5.2b (*Dobin et al., 2013*). Subsequently, the sequencing data was mapped to the indexed genome using the BD Rhapsody single-nucleus pipeline v1.9, employing default parameters.

Each data matrix was converted to a SingleCellExperiment data format, and empty barcodes were removed using the emptyDrops function of DropletUtils v3.14 (*Lun et al., 2019*). Then, we converted and processed the data in Seurat v3 (*Stuart et al., 2019*). We removed cells that had <100 or >2500 genes and <100 or >6000 unique molecular identifiers (UMIs). We also removed genes that had <10 UMIs in each data matrix. Then, we log-normalised the data and used Doublet-Finder v2.0 (*McGinnis et al., 2019*) to remove potential doublet, assuming a 7.5% doublet formation rate. The numbers of retained nuclei were 9717, 21,614, and 28,928 for Fanmao, starvation, and reconstitution data, respectively (*Supplementary file 1t*). We used the top 3000 highly variable genes for principal component analysis (PCA) and a reciprocal PCA approach to integrating the three datasets (*Stuart et al., 2019*). The first 40 principal components (PCs) were used for UMAP dimensional reduction and following clustering (*Figure 1—figure supplement 3*). We employed an empirical parameter – a resolution of 0.2 – in the FindClusters function, utilising the original Louvain algorithm. To identify unique marker genes associated with each cluster, we utilised the FindAllMarkers function from the Seurat package. This analysis employed the Wilcoxon rank-sum test, focusing on genes that demonstrated a minimum 0.3-fold difference between two groups of cells. The annotation of cell types relied on the reference of previously published marker genes. In instances where clusters exhibited marker genes that could not be associated with known cell types, we pursued the validation process through WISH (elaborated below), discerning cell types based on gene expression patterns and morphological characteristics. When WISH displayed uniform expression of marker genes from different clusters within the same cell type, we consolidated those clusters into a single cell type. We assigned cells to 14 reliable cell types (*Figure 1—figure*

*supplements 3 and 4*). *Supplementary file 2* presents the counting matrix, and *Supplementary file 3* presents the average expression of each gene per cell type. To evaluate the stability of each cell type, we implemented a bootstrap sampling and clustering strategy comprising 100 iterations using cells combined from all three samples and individual samples (*Singh and Zhai, 2022*). The determination of marker genes per cell type followed the previously described methodology. The identified marker genes were subsequently utilised in KEGG enrichment analysis, employing ClusterProfiler v4.2 (*Wu et al., 2021*).

## Cell trajectory analysis

We conducted slingshot trajectory analyses using Slingshot v2.2.1 (*Street et al., 2018*) for a subset of clusters (intercalary cells, dorsal-end proliferation cells, ventral-end proliferation cells, mucus cells, basal membrane cells 1, basal membrane cells 2, and bacteriocytes) to explore the developmental trajectory of cells, assuming that all these cells were developed from the same precursor ('posterior-end budding zone'). We performed PCA for the subsampled data set and conducted dimensional reduction using phateR v1.0.7 for slingshot analyses. We used the Potential of Heat-diffusion for Affinity-based Trajectory Embedding (PHATE) because it could better reveal developmental branches than other tools (*Moon et al., 2019*).

We also examined the effect of deep-sea transplant experiments on shaping gene expression patterns by comparing the expression levels amongst the three different states of a given cluster. We conducted Monocle analyses using Monocle2 and Monocle 3 in R environment (*Qiu et al., 2017*; *Cao et al., 2019*). This comparison was done for the bacteriocytes. The biased number of cells per state could affect the results of the dimensional reduction and calculation of marker genes, and the sequenced nucleus per state was unbalanced; therefore, we first downsampled the cells per cluster per state to a maximum of 1000 nuclei per cell type. Thereafter, we performed PCA and dimensional reduction using UMAP and PHATE, calculated marker genes and conducted slingshot trajectory and KEGG enrichment analyses as mentioned above for each cell type.

## Phylogenetic estimation

Phylogenetic estimation was conducted for E3 ubiquitin ligase RNF213 genes. We downloaded RNF213 genes from GenBank for representative vertebrate and invertebrate species across the tree of life. The amino acid sequences were aligned with the *G. platifrons* sequences annotated as E3 ubiquitin ligase (Bpl_scaf_25983-1.26 and Bpl_scaf_1894-1.45) using MAFFT v7.450. The alignment was used to estimate a maximum-likelihood best gene tree and calculate bootstrap values on each node using RAxML v8.2.

## *G. platifrons* gill fixation and storage

The gill tissues of *G. platifrons* collected from Fanmao site were dissected within minutes after the ROV, and samples were retrieved onboard *R/V Kexue*. The gill tissues were briefly washed with ice-cold filtered and autoclaved in situ seawater (FAISW) and then fixed in 4% PFA prepared in FAISW at 4°C overnight. The gill tissues were washed three times with ice-cold PBST, dehydrated in 100% methanol, and stored at –20°C until use.

## Synthesise probes for mRNA in situ hybridisations

For WISH and double FISH analyses, the DNA fragments (~1000 bp) of the targeted genes were first PCR amplified with gene-specific primers (GSPs) pairs using *G. platifrons* gill cDNA as template (the sequences of targeted genes and GSPs are provided in *Supplementary file 4*). The amplified fragments were ligated into the pMD18-simpleT vector (Takara) and transformed into *Escherichia coli*. Individual colonies were picked up and their plasmids were sequenced to confirm the inserts. The templates for in vitro mRNA transcription were amplified using T7 forward GSP (sense probe control) or Sp6 reversed GSPs (antisense probe) combined with either forward or reversed gene-specific primer. Labelled probes and control probes were generated using digoxigenin (DIG)–12-UTP (Roche) or fluorescein-12-UTP (Roche) according to the protocol described by *Thisse and Thisse, 2008* with Sp6 and T7 RNA polymerase, respectively.

## Paraffin embedding and double fluorescent in situ hybridisation

The methanal-dehydrated gill slices were incubated in 100% ethanol, a 1:1 mixture of 100% ethanol and xylene and xylene twice for 1 hr each at room temperature (RT). The samples were embedded by incubating in Paraplast Plus (Sigma-Aldrich) for 2 hr at 65°C and then cooled down to RT. Sections with 5 µM thickness were cut using a microtome (Leica).

For double FISH, sections were dewaxed by incubating in xylene twice, a 1:1 mixture of 100% ethanol and xylene, 100% ethanol twice, 95% ethanol, 85% ethanol, and 75% ethanol for 15 min each at RT. The sections were washed with PBST three times for 10 min each and then permeabilised by 2 µg/mL proteinase K (NEB) in PBST for 15 min at RT. Post-digestion fixation was conducted by incubating the sections in 4% PFA in PBST for 30 min at RT. The sections were washed three times with PBST for 15 min each. Pre-hybridisation was conducted by incubating the sections in HM for 1 hr at 55°C. Then, the in situ hybridisation was performed by incubating the sections in ~0.5 ng/µL fluorescein (Roche)-labelled probed prepared in fresh HM overnight at 55°C. The sections were washed three times with 2× saline–sodium citrate (SSC) for 15 min each at 55°C, cooled down to RT, and washed three times with PBST.

A second-round hybridisation was conducted on the DIG-labelled oligonucleotides to label the symbiont. The slices were hybridised for 1 hr at 46°C with 100 ng DIG-labelled *G. platifrons* symbiont-specific probe in FISH buffer (0.9 M NaCl, 0.02 M Tris–HCl, 0.01% sodium dodecyl sulphate [SDS], and 30% formamide). The slices were then washed with FISH washing buffer (0.1 M NaCl, 0.02 M Tris–HCl, 0.01% SDS, and 5 mM EDTA) three times at 5 min each at 48°C. The slices were washed with PBST three times and then blocked with blocking buffer (2.5% sheep serum and 2% BSA in sheep serum) for 1 hr at RT.

The slices were then incubated with 1:1000 diluted anti-fluorescein–peroxidase (POD) (Roche) overnight at 4°C, then washed six times with PBST for 15 min each and three times with TNT buffer (100 mM Tris–HCl, pH 7.5; 100 mM NaCl; 0.1% Tween 20) for 15 min each. Afterwards, the fluorescent signal of the *G. platifrons* gene expression pattern was developed using the TSA fluorescein kit (Akoya Biosciences) according to the manufacturer's protocol. The slices were washed three times, and the remaining POD activity was quenched by incubation in 1% hydrogen peroxide solution for 1 hr at RT. Then, the slices were washed three times with PBS, blocked with blocking buffer for 30 min at RT, incubated with 1:2500 diluted anti-DIG–POD (Roche) for 2 hr at RT, and washed with PBST six times and TNT three times. The symbiont FISH signal was developed using the TSA Cy3 kit (Akoya Biosciences). Finally, the slices were washed with PBST, stained with DAPI, and mounted with ProLong Diamond Antifade Mountant (Thermo Fisher).

## Whole-mount in situ hybridisation

For WISH, the connective region at the end of the W-shaped gill filament was cut off, and each gill slice was carefully peeled off with fine-tip tweezers. We dissected gill tissues from five individual mussels and pooled all the gill slices together. The gill slices were then rehydrated in 75, 50, and 25% methanol-PBST (1× PBS with 0.1% Tween 20) for 15 min each, followed by 3 × 5 min PBST washes. The gill slices were then permeabilised with 2 µg/mL proteinase K in PBST for 30 min at 37°C. Post-digestion fixation was conducted by fixing the gill slices with 4% PFA in PBST for 30 min at RT. After 3 × 5 min PBST wash to remove the residual fixative, the gill slices were pre-hybridised with a hybridisation mix (HM; containing 50% formamide, 5× SSC, 0.1% Tween 20, 10 µg/mL heparin, 500 µg/mL yeast tRNA) for 1 hr at 65°C. For each hybridisation, 5–10 gill slices were added to 400 µL of fresh HM containing ~0.5 ng/µL DIG-labelled probe. Hybridisation was conducted in a 55°C shaking water bath overnight. Post-hybridisation washes were performed according to the following steps: the gill slices were first washed 3 × 15 min with hybridisation washing buffer (50% formamide, 5× SSC, 0.1% Tween 20), followed by 3 × 15 min 2× SSC with 0.1% Tween 20 and 3 × 15 min 0.2× SSC with 0.1% Tween 20. The washings were also conducted in a shaking water batch, and all the washing buffers were pre-heated to 55°C. The samples were washed three times with PBST and then blocked in blocking buffer (2.5% sheep serum, 2% BSA in PBST) for 1 hr at RT. Each sample was incubated with 1:10,000 diluted anti-DIG-AP antibody (Roche) at 4°C overnight. The samples were incubated with the antibody, then washed for 6 × 15 min with PBST, followed by washing in 3 × 15 min alkaline Tris buffer (100 mM Tris–HCl, pH 9.5; 100 mM NaCl; 50 mM MgCl₂). The samples were incubated in nitro blue tetrazolium/5-bromo-4-chloro-3-indolyl phosphate staining solution (Sangon). After the desired

expression pattern was revealed, the staining reaction was stopped by 3 × 15 min PBST–EDTA wash (PBST, 1 mM ETDA). The gill slices were cleared by incubating in 100% glycerol overnight at 4°C and then mounted on glass slides. The results of control hybridisations (with sense probes) are provided in *Figure 2—figure supplement 1*, *Figure 3—figure supplement 4*, *Figure 4—figure supplement 2*, and *Figure 5—figure supplement 1*. WISH analyses were repeated with another batch of *G. platifrons* gill slices samples collected during the *R/V* 'Kexue' 2017 South China Sea cold seep cruise to confirm the consistency in expression patterns.

## Microscopy imaging

All the WISH samples and whole-mount 16S FISH images were observed and imaged with a Nikon Eclipse Ni microscope with a DS-Ri2 camera. The double FISH slides were imaged with a Zeiss LSM710 confocal microscope.

## Electron microscopy analysis

The gill slices of the *G. platifrons* were dissected and fixed in electron microscopy fixative (2.5% glutaraldehyde and 2% PFA) at 4°C. For SEM analysis, the samples were dehydrated in a graded ethanol series and then dried at the critical point. The samples were then coated with gold (sputter/ carbon Thread, EM ACE200) and observed under a scanning electron microscope (VEGA3, Tescan). For TEM analysis, the samples were rinsed with double-distilled water, post-fixed with 1% osmium tetroxide, and then washed with double-distilled water. The samples were then rinsed, dehydrated, and embedded in Ep812 resin. Ultra-thin sections were obtained with an ultramicrotome (70 nm thickness, Reichert-Jung Ultracut E). The sections were then double-stained with lead citrate and uranyl acetate. The cells were observed under a transmission electron microscope (JEM1200, Jeol) operated under 100 kV.

## Acknowledgements

This study was supported by the Science and Technology Innovation Project of Laoshan Laboratory (Project Number No. LSKJ202203104), the National Natural Science Foundation of China (Grant No. 42030407), Southern Marine Science and Engineering Guangdong Laboratory (Guangzhou) (HJ202101, SMSEGL24SC01), Major Project of Basic and Applied Basic Research of Guangdong Province (2019B030302004), the Research Grants Council of Hong Kong (C2013-22G and 16101822), and the Guangdong Natural Science Funds for Distinguished Young Scholar (2022B1515020033). We appreciate all the assistance provided by the crew of *R/V* '*Kexue*' and the operation team of ROV 'Faxian'.

---

## Additional information

### Funding

| Funder | Grant reference number | Author |
|---|---|---|
| Science and Technology Innovation Project of Laoshan Laboratory | LSKJ202203104 | Hao Wang |
| National Natural Science Foundation of China | 42030407 | Chaolun Li |
| Southern Marine Science and Engineering Guangdong Laboratory | HJ202101 | Pei-Yuan Qian |
| Southern Marine Science and Engineering Guangdong Laboratory | SMSEGL24SC01 | Pei-Yuan Qian |
| Major Project of Basic and Applied Basic Research of Guangdong Province | 2019B030302004 | Pei-Yuan Qian |

| Funder | Grant reference number | Author |
|---|---|---|
| Research Grants Council of Hong Kong | C2013-22G | Pei-Yuan Qian |
| Research Grants Council of Hong Kong | 16101822 | Pei-Yuan Qian |
| Guangdong Natural Science Funds for Distinguished Young Scholar | 2022B1515020033 | Kai He |

The funders had no role in study design, data collection and interpretation, or the decision to submit the work for publication.

## Author contributions

Hao Wang, Conceptualization, Resources, Formal analysis, Validation, Investigation, Visualization, Methodology, Writing – original draft, Project administration, Writing – review and editing; Kai He, Conceptualization, Data curation, Software, Formal analysis, Supervision, Validation, Visualization, Methodology, Writing – original draft, Writing – review and editing; Huan Zhang, Lei Cao, Jing Li, Formal analysis, Validation, Investigation, Visualization, Methodology; Quanyong Zhang, Data curation, Investigation, Methodology; Zhaoshan Zhong, Li Zhou, Data curation, Formal analysis, Validation, Investigation, Visualization, Methodology; Hao Chen, Data curation, Formal analysis, Validation, Visualization, Methodology; Chao Lian, Resources, Formal analysis, Validation, Investigation, Visualization, Methodology; Minxiao Wang, Resources, Data curation, Formal analysis, Validation, Investigation, Visualization, Methodology; Kai Chen, Conceptualization, Resources, Data curation, Software, Formal analysis, Supervision, Validation, Investigation, Visualization, Methodology, Writing – original draft, Project administration, Writing – review and editing; Pei-Yuan Qian, Conceptualization, Resources, Data curation, Supervision, Funding acquisition, Investigation, Writing – original draft, Project administration, Writing – review and editing; Chaolun Li, Conceptualization, Resources, Data curation, Formal analysis, Supervision, Funding acquisition, Investigation, Project administration

## Author ORCIDs

Hao Wang https://orcid.org/0000-0002-9435-5501
Zhaoshan Zhong https://orcid.org/0000-0003-1643-5407
Hao Chen https://orcid.org/0000-0001-6697-0809
Pei-Yuan Qian https://orcid.org/0000-0003-4074-9078

Reviewer #1 (Public Review): https://doi.org/10.7554/eLife.88294.4.sa1
Reviewer #2 (Public Review): https://doi.org/10.7554/eLife.88294.4.sa2
Reviewer #3 (Public Review): https://doi.org/10.7554/eLife.88294.4.sa3
Author response https://doi.org/10.7554/eLife.88294.4.sa4

# Additional files

## Supplementary files

Supplementary file 1. The supplementary tables. (**a**) Summary statistics of single-cell data information, including numbers and percentages of cells and numbers of available genes per cluster in each sample individually and in all three pieces combined. (**b**) Identified cell markers of the supportive cells, including the inter lamina cell, the basal membrane cell 1, the basal membrane cell 2, and the mucus cell. (**c**) Identified cell markers of the ciliary cells, including the apical ridge ciliary cell, the food grove ciliary cell, the lateral ciliary cell, the smooth muscle cell, and the intercalary cell. (**d**) Identified cell markers of the proliferation cells, including the posterior end budding zone, the dorsal end proliferation cell, and the ventral end proliferation cell. (**e**) Identified cell markers of the bacteriocyte. (**f**) Cell cluster-specific differentially expressed genes among the three deep-sea in situ treatments, the inter lamina cell. (**g**) Cell cluster-specific differentially expressed genes among the three deep-sea in situ treatments, the basal membrane cell 1. (**h**) Cell cluster-specific differentially expressed genes among the three deep-sea in situ treatments, the basal membrane cell 2. (**i**) Cell cluster-specific differentially expressed genes among the three deep-sea in situ treatments, the

mucus cell. (**j**) Cell cluster-specific differentially expressed genes among the three deep-sea in situ treatments, the apical ridge ciliary cell. (**k**) Cell cluster-specific differentially expressed genes among the three deep-sea in situ treatments, the food grove ciliary cell. (**l**) Cell cluster-specific differentially expressed genes among the three deep-sea in situ treatments, the lateral ciliary cell. (**m**) Cell cluster-specific differentially expressed genes among the three deep-sea in situ treatments, the smooth muscle cell. (**n**) Cell cluster-specific differentially expressed genes among the three deep-sea in situ treatments, the intercalary cell. (**o**) Cell cluster-specific differentially expressed genes among the three deep-sea in situ treatments, the posterior end budding zone. (**p**) Cell cluster-specific differentially expressed genes among the three deep-sea in situ treatments, the dorsal end proliferation cell. (**q**) Cell cluster-specific differentially expressed genes among the three deep-sea in situ treatments, the ventral end proliferation cell. (**r**) Cell cluster-specific differentially expressed genes among the three deep-sea in situ treatments, the bacteriocytes. (**s**) Euclid distances between cell types in three different treatments. The distances were computed based on the centroid coordinates of each cell type in each condition and are shown in *Figure 6B*. (**t**) Quality control for each sample sequenced using the BD Rhapsody platform.

Supplementary file 2. snRNA-seq expression counting matrix.

Supplementary file 3. Average expression of each gene per cell type.

Supplementary file 4. The sequences of targeted genes and gene-specific primers.

MDAR checklist

### Data availability

The sequencing datasets were submitted to the National Centre for Biotechnology Information Sequence Read Archive database (Bioproject: PRJNA755857).

The following dataset was generated:

| Author(s) | Year | Dataset title | Dataset URL | Database and Identifier |
|---|---|---|---|---|
| Wang H | 2021 | Gigantidas platifrons gill snRNA-seq | https://www.ncbi.nlm.nih.gov/bioproject/?term=PRJNA755857 | NCBI BioProject, PRJNA755857 |

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
