## [Editor Report · eLife assessment]

This study provides an **important** cell-type atlas of the gill of the mussel *Gigantidas platifrons* using a single-nucleus RNA-seq dataset, a resource for the community of scientists studying deep-sea physiology and metabolism and intracellular host–symbiont relationships. The evidence supporting the conclusions is **convincing** with high-quality single-nucleus RNA sequencing and transplant experiments. This work will be of broad relevance for scientists interested in host–symbiont relationships across ecosystems.

---

## [Referee Report · Reviewer #1 (Public Review)]

Wang, He et al have constructed a comprehensive single nucleus atlas for the gills of the deep sea Bathymodioline mussels, which possess intracellular symbionts that provide a key source of carbon and allow them to live in these extreme environments. They provide annotations of the different cell states within the gills, shedding light on how multiple cell types cooperate to give rise to the emergent functions of the composite tissues and the gills as a whole. They pay special attention to characterizing the bacteriocyte cell populations and identifying sets of genes that may play a role in their interaction with the symbiotes.

Wang, He et al sample mussels from 3 different environments: animals from their native methane rich environment, animals transplanted to a methane-poor environment to induce starvation and animals that have been starved in the methane-poor environment and then moved back to the methane-rich environment. They demonstrated that starvation had the biggest impact on bacteriocyte transcriptomes. They hypothesize that the up-regulation of genes associated with lysosomal digestion leads to the digestion of the intracellular symbiont during starvation, while the non-starved and reacclimated groups more readily harvest the nutrients from symbiotes without destroying them. Further work exploring the differences in symbiote populations between ecological conditions will further elucidate the dynamic relationship between host and symbiote. This will help disentangle specific changes in transcriptomic state that are due to their changing interactions with the symbiotes from changes associated with other environmental factors.

This paper makes available a high quality dataset that is of interest to many disciplines of biology. The unique qualities of this non-model organism and collection of conditions sampled make it of special interest to those studying deep sea adaptation, the impact of environmental perturbation on Bathymodioline mussels populations, and intracellular symbiotes. The authors also use a diverse array of tools to explore and validate their data.

---

## [Referee Report · Reviewer #2 (Public Review)]

Wang, He et al. shed insight into the molecular mechanisms of deep-sea chemosymbiosis at the single-cell level. They do so by producing a comprehensive cell atlas of the gill of Gigantidas platifrons, a chemosymbiotic mussel that dominates the deep-sea ecosystem. They uncover novel cell types and find that the gene expression of bacteriocytes, the symbiont-hosting cells, supports two hypotheses of host-symbiont interactions: the "farming" pathway, where symbionts are directly digested, and the "milking" pathway, where nutrients released by the symbionts are used by the host. They perform an in situ transplantation experiment in the deep sea and reveal transitional changes in gene expression that support a model where starvation stress induces bacteriocytes to "farm" their symbionts, while recovery leads to the restoration of the "farming" and "milking" pathways.

A major strength of this study includes the successful application of advanced single nucleus techniques to a non-model, deep sea organism that remains challenging to sample. I also applaud the authors for performing an in situ transplantation experiment in a deep sea environment. From gene expression profiles, the authors deftly provide a rich functional description of G. platifrons cell types that is well-contextualized within the unique biology of chemosymbiosis. These findings offer significant insight into the molecular mechanisms of deep-sea host-symbiont ecology, and will serve as a valuable resource for future studies into the striking biology of G. platifrons.

The authors' conclusions are generally well-supported by their results. However, I recognize that the difficulty of obtaining deep-sea specimens may have impacted experimental design and no replicates were sampled.

It is notable that the Fanmao cells were much more sparsely sampled. It appears that fewer cells were sequenced, resulting in the Starvation and Reconstitution conditions having 2-3x more cells after doublet filtering. These discrepancies also are reflected in the proportion of cells that survived QC, suggesting a distinction in quality or approach. However, the authors provide clear and sufficient evidence via bootstrapping that batch effects between the three samples are negligible. While batch effect does not appear to have affected gene expression profiles, the proportion of cell types may remain sensitive to sampling techniques, and thus interpretation of Fig. S12 must be approached with caution.

---

## [Referee Report · Reviewer #3 (Public Review)]

Wang et al. explored the unique biology of the deep-sea mussel Gigantidas platifrons to understand fundamental principles of animal-symbiont relationships. They used single-nucleus RNA sequencing and validation and visualization of many of the important cellular and molecular players that allow these organisms to survive in the deep-sea. They demonstrate that a diversity of cell types that support the structure and function of the gill including bacteriocytes, specialized epithelial cells that host sulfur-oxidizing or methane-oxidizing symbionts as well as a suite of other cell types including supportive cells, ciliary, and smooth muscle cells. By performing experiments of transplanting mussels from one habitat which is rich in methane to methane-limited environments, the authors showed that starved mussels may consume endosymbionts versus in methane-rich environments upregulated genes involved in glutamate synthesis. These data add to the growing body of literature that organisms control their endosymbionts in response to environmental change.

The conclusions of the data are well supported. The authors adapted a technique that would have been technically impossible in their field environment by preserving the tissue and then performing nuclear isolation after the fact. The use of single-nucleus sequencing opens the possibility of new cellular and molecular biology that is not possible to study in the field. Additionally, the in-situ data (both WISH and FISH) are high-quality and easy to interpret. The use of cell-type-specific markers along with a symbiont-specific probe was effective. Finally, the SEM and TEM were used convincingly for specific purposes in the case of showing the cilia that may support water movement.

The one particular area for future exploration surrounds the concept of a proliferative progenitor population within the gills. The authors recover molecular markers for these putative populations and additional future work will uncover if these are indeed proliferative cells that contribute to symbiont colonization.

Overall the significance of this work is identifying the relationship between symbionts and bacteriocytes and how these host bacteriocytes modulate their gene expression in response to environmental change. It will be interesting to see how similar or different these data are across animal phyla. For instance, the work of symbiosis in cnidarians may converge on similar principles of there may be independent ways in which organisms have been able to solve these problems.

---

## [Author Response]

The following is the authors’ response to the previous reviews.

**Public Reviews:**

**Reviewer #1 (Public Review):**
Wang, He et al have constructed comprehensive single nucleus atlas for the gills of the deep sea Bathymodioline mussels, which possess intracellular symbionts that provide a key source of carbon and allow them to live in these extreme environments. They provide annotations of the different cell states within the gills, shedding light on how multiple cell types cooperate to give rise to the emergent functions of the composite tissues and the gills as a whole. They pay special attention to characterizing the bacteriocyte cell populations and identifying sets of genes that may play a role in their interaction with the symbiotes.Wang, He et al sample mussels from 3 different environments: animals from their native methane rich environment, animals transplanted to a methane-poor environment to induce starvation and animals that have been starved in the methane-poor environment and then moved back to the methane-rich environment. They demonstrated that starvation had the biggest impact on bacteriocyte transcriptomes. They hypothesize that the up-regulation of genes associated with lysosomal digestion leads to the digestion of the intracellular symbiont during starvation, while the non-starved and reacclimated groups more readily harvest the nutrients from symbiotes without destroying them. Further work exploring the differences in symbiote populations between ecological conditions will further elucidate the dynamic relationship between host and symbiote. This will help disentangle specific changes in transcriptomic state that are due to their changing interactions with the symbiotes from changes associated with other environmental factors.This paper makes available a high quality dataset that is of interest to many disciplines of biology. The unique qualities of this non-model organism and collection of conditions sampled make it of special interest to those studying deep sea adaptation, the impact of environmental perturbation on Bathymodioline mussels populations, and intracellular symbiotes. The authors also use a diverse array of tools to explore and validate their data.
**Reviewer #2 (Public Review):**
Wang, He et al. shed insight into the molecular mechanisms of deep-sea chemosymbiosis at the single-cell level. They do so by producing a comprehensive cell atlas of the gill of Gigantidas platifrons, a chemosymbiotic mussel that dominates the deep-sea ecosystem. They uncover novel cell types and find that the gene expression of bacteriocytes, the symbiont-hosting cells, supports two hypotheses of host-symbiont interactions: the "farming" pathway, where symbionts are directly digested, and the "milking" pathway, where nutrients released by the symbionts are used by the host. They perform an in situ transplantation experiment in the deep sea and reveal transitional changes in gene expression that support a model where starvation stress induces bacteriocytes to "farm" their symbionts, while recovery leads to the restoration of the "farming" and "milking" pathways.A major strength of this study includes the successful application of advanced single nucleus techniques to a non-model, deep sea organism that remains challenging to sample. I also applaud the authors for performing an in situ transplantation experiment in a deep sea environment. From gene expression profiles, the authors deftly provide a rich functional description of G. platifrons cell types that is well-contextualized within the unique biology of chemosymbiosis. These findings offer significant insight into the molecular mechanisms of deep-sea host-symbiont ecology, and will serve as a valuable resource for future studies into the striking biology of G. platifrons.The authors' conclusions are generally well-supported by their results. However, I recognize that the difficulty of obtaining deep-sea specimens may have impacted experimental design and no replicates were sampled.It is notable that the Fanmao cells were much more sparsely sampled. It appears that fewer cells were sequenced, resulting in the Starvation and Reconstitution conditions having 2-3x more cells after doublet filtering. These discrepancies also are reflected in the proportion of cells that survived QC, suggesting a distinction in quality or approach. However, the authors provide clear and sufficient evidence via bootstrapping that batch effects between the three samples are negligible. While batch effect does not appear to have affected gene expression profiles, the proportion of cell types may remain sensitive to sampling techniques, and thus interpretation of Fig. S12 must be approached with caution.
**Reviewer #3 (Public Review):**
Wang et al. explored the unique biology of the deep-sea mussel Gigantidas platifrons to understand fundamental principles of animal-symbiont relationships. They used single-nucleus RNA sequencing and validation and visualization of many of the important cellular and molecular players that allow these organisms to survive in the deep-sea. They demonstrate that a diversity of cell types that support the structure and function of the gill including bacteriocytes, specialized epithelial cells that host sulfur-oxidizing or methane-oxidizing symbionts as well as a suite of other cell types including supportive cells, ciliary, and smooth muscle cells. By performing experiments of transplanting mussels from one habitat which is rich in methane to methane-limited environments, the authors showed that starved mussels may consume endosymbionts versus in methane-rich environments upregulated genes involved in glutamate synthesis. These data add to the growing body of literature that organisms control their endosymbionts in response to environmental change.The conclusions of the data are well supported. The authors adapted a technique that would have been technically impossible in their field environment by preserving the tissue and then performing nuclear isolation after the fact. The use of single-nucleus sequencing opens the possibility of new cellular and molecular biology that is not possible to study in the field. Additionally, the in-situ data (both WISH and FISH) are high-quality and easy to interpret. The use of cell-type-specific markers along with a symbiont-specific probe was effective. Finally, the SEM and TEM were used convincingly for specific purposes in the case of showing the cilia that may support water movement.The one particular area for future exploration surrounds the concept of a proliferative progenitor population within the gills. The authors recover molecular markers for these putative populations and additional future work will uncover if these are indeed proliferative cells contribute to symbiont colonization.Overall the significance of this work is identifying the relationship between symbionts and bacteriocytes and how these host bacteriocytes modulate their gene expression in response to environmental change. It will be interesting to see how similar or different these data are across animal phyla. For instance, the work of symbiosis in cnidarians may converge on similar principles of there may be independent ways in which organisms have been able to solve these problems.

We extend our sincere gratitude to all the reviewers for their positive comments and kind words. We highly value the substantial efforts they made in helping us improve and enhance our manuscript. Additionally, we appreciate the reviewers for pointing out the limitations of our current study, which will guide us in improving our future researches.

**Recommendations for the authors:**

**Reviewer #1 (Recommendations For The Authors):**
This study system is so interesting and this is a truly unique and exciting dataset. Most of my suggestions are aimed at improving readability and making it more accessible for a broader audience, since I predict many fields will find it interesting.Line 60: which species of mussel? Is this the same one?

We appreciate the comments from the reviewer. The reference here is to deep-sea bathymodiolin mussels, which, in most cases, possess enlarged gill filaments that accommodate symbionts.

Line 237-230: citation of previous findings missing

We appreciate the comments from the reviewer. After carefully reviewing these paragraphs, we believe that all the previous findings have now been properly cited.

Line 256: it might be a good idea to give a brief description of what slingshot analysis is here

We appreciate the comments from the reviewer. We have revise the corresponding part of our manuscript to make it clear.

This parts of manscript now reads: “We performed Slingshot analysis, which uses a cluster-based minimum spanning tree (MST) and a smoothed principal curve to determine the developmental path of cell clusters. The re-sult shows that the PEBZCs might be the origin of all gill epithelial cells, including the other two proliferation cells (VEPC and DEPC) and bacteriocytes (Supplementary Fig. S6).” Line 203-207 of the revised manscript.

Line 289: Wording is a bit confusing- what is meant by morphological analysis?

We acknowledge that our wording might be a bit confusing here. We are referring to the TEM ultrastructural analysis. Therefore, we have changed “morphological analysis” to “ultrastructural analysis.” Line231 in the revised manuscript.

Line 351-354: how did you calculate distances? How many dimensions were used?

We calculated the centroid coordinates for each cell type in each state on the 2-dimensional UMAP plot (Fig. 6A). Then, for each cell type, we determined the Euclidean distance between the centroid coordinates of each pair of states. We have revised the manuscript with this more detailed description. Line 292-295 of revised manuscript.

Line 462: identify -> identified

We apologize for our mistake and appreciate the reviewer’s kind assistance with proofreading. The typo has been corrected in the new version. Line396 of the revised manscript.

Line 509: what does the size of the dot represent?

In this context, the color and intensity of each dot represent a specific gene’s expression level in the single-cell cluster. The dot size is universal and therefore does not convey a specific meaning.

Fig 3A: What is the blue cluster highlighted?

We apologize for our mistake. The label for the teal box was missed. We have corrected our mistake in the revised manuscript.

Fig 3K: Wording in key is confusing.

We have modified our description of Fiugre 3K in the figure legneds. Now it reads: “Schematic of water flow agitated by different ciliary cell types. The color of arrowheads corresponds to water flow potentially influenced by specific types of cilia, as indicated by their color code in Figure 3A.” Line462-464 in the revised manscript.

Fig 5B: which population of mussels was used to take these images?

These mussels from “Fanmao” (methane rich) site were used to take these images. We have revised our material and methods to make it clear. Line602-603 of the revised manuscript.

Fig 5E,5G,5H: panels not referenced in text

We apologize for our mistake and appreciate the reviewer’s thorough reading. This error has been corrected in the new version of the manuscript. Line233 of the revised manuscript.

**Reviewer #2 (Recommendations For The Authors):**
Minor comments:Fig. 3A - the teal box in the legend lacks a label

We apologize for our mistake. The label for the teal box was missed. We have corrected our mistake in the

**Reviewer #3 (Recommendations For The Authors):**
My enthusiasm for the manuscript remains high and I appreciate the authors care in responding to the various reviewer questions and concerns.In regards to the cell proliferation results, I have modified my public review and look forward to your future work in this area. The data for both pHistone H3 and anti PCNA are compelling!One typo I did catch occurs on line 520. I believe you meant to say "outer" not "otter."

We apologize for our mistake and appreciate the reviewer’s kind assistance with proofreading. The typo has been corrected in the new version.